# HypRAG: Hyperbolic Dense Retrieval for Retrieval Augmented Generation

**Hiren Madhu** [1]  **Ngoc Bui** [1]  **Ali Maatouk** [1]  **Leandros Tassiulas** [1]  **Smita Krishnaswamy** [1]  **Menglin Yang** [2]
**Sukanta Ganguly** [3]  **Kiran Srinivasan** [3]  **Rex Ying** [1]

## Abstract

Embedding geometry plays a fundamental role in retrieval quality, yet dense retrievers for retrieval-augmented generation (RAG) remain largely confined to Euclidean space. However, natural language exhibits hierarchical structure from broad topics to specific entities that Euclidean embeddings fail to preserve, causing semantically distant documents to appear spuriously similar and increasing hallucination risk. To address these limitations, we introduce hyperbolic dense retrieval, developing two model variants in the Lorentz model of hyperbolic space: HyTE-FH, a fully hyperbolic transformer, and HyTE-H, a hybrid architecture projecting pre-trained Euclidean embeddings into hyperbolic space. To prevent representational collapse during sequence aggregation, we introduce the Outward Einstein Midpoint, a geometry-aware pooling operator that provably preserves hierarchical structure. On MTEB, HyTE-FH outperforms equivalent Euclidean baselines, while on RAG-Bench, HyTE-H achieves up to 29% gains over Euclidean baselines in context relevance and answer relevance using substantially smaller models than current state-of-the-art retrievers. Our analysis also reveals that hyperbolic representations encode document specificity through norm-based separation—with over 20% radial increase from general to specific concepts—a property absent in Euclidean embeddings, underscoring the critical role of geometric inductive bias in faithful RAG systems. The code is available at: https://github.com/Graph-and-Geometric-Learning/HypRAG.

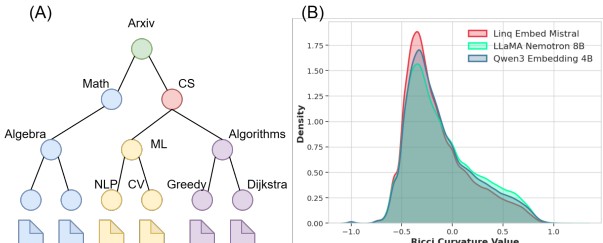

*Figure 1.* **Hierarchies in Text.** (A) Documents naturally organize into branching hierarchies where general topics spawn increasingly specific subtopics. Euclidean spaces distort such hierarchies due to crowding effects, while hyperbolic geometry preserves hierarchical relationships through exponential volume growth. (B) Ricci curvature analysis of document embeddings from strong baselines reveals predominantly negative curvature, indicating tree-like semantic structure.

## 1. Introduction

Dense retrieval forms the backbone of retrieval-augmented generation (RAG) systems (Lewis et al., 2020; Fan et al., 2024), where embedding quality directly determines whether generated responses are grounded in evidence or hallucinated. By retrieving relevant documents and conditioning generation on this context, RAG systems produce responses that are more attributable and aligned with verifiable sources (Ni et al., 2025). Yet, despite advances in retrieval architectures, current systems continue to rely on Euclidean embeddings, a choice inherited from standard neural networks rather than from language structure itself.

Natural language inherently exhibits strong hierarchical organization (He et al., 2025b; Robinson et al., 2024), with semantic structure giving rise to locally tree-like neighborhoods. Euclidean spaces struggle to represent such branching hierarchies due to polynomial volume growth (He et al., 2025b), introducing shortcuts between hierarchically distinct regions that distort semantic relationships. In retrieval settings, these distortions can cause semantically distant documents to appear spuriously similar (Radovanovic et al., 2010; Bogolin et al., 2022), degrading retrieval precision (Reimers & Gurevych, 2021): a query about a specific subtopic may retrieve documents from sibling or parent categories that share similarity but lack the required specificity.

To further see why geometry matters for retrieval, consider a query about transformer attention mechanisms (Figure 1A).

---
[1]Yale University, USA [2]Hong Kong University of Science and Technology (Guangzhou), China [3]NetApp, USA. Correspondence to: Hiren Madhu <hiren.madhu@yale.edu>.

*Proceedings of the 43rd International Conference on Machine Learning*, Seoul, South Korea. PMLR 306, 2026. Copyright 2026 by the author(s).

Relevant documents form a natural hierarchy—from general concepts like NLP, to transformers, to specific components like multi-head attention—inducing tree-like semantic structure. Euclidean embeddings struggle to preserve this organization: representing both broad topics and specialized descendants forces a trade-off between semantic proximity and fine-grained separation, causing neighborhood crowding and distortion. Hyperbolic geometry resolves this tension through exponential volume growth, allowing general concepts to remain compact while specific documents spread outward. To test whether such structure appears empirically, we analyze Ollivier–Ricci curvature (Ni et al., 2019)—a measure of local geometry where negative values indicate tree-like branching—on graphs built from MS MARCO document embeddings (Bajaj et al., 2016). Across several strong models (Linq Embed Mistral, LLaMA Nemotron 8B, Qwen3 Embedding 4B), curvature distributions are predominantly negative (Figure 1B), providing empirical evidence that retrieval-relevant embeddings exhibit intrinsic hyperbolic structure and motivating hyperbolic geometry as a natural inductive bias for dense retrieval.

Recent work has begun exploring hyperbolic geometry for language modeling and RAG systems, though with different focus areas. HELM (He et al., 2025a) introduces a family of hyperbolic language models that operate entirely in hyperbolic space, but these models target text generation rather than retrieval. In the RAG setting, HyperbolicRAG (Cao et al., 2025) projects embeddings into the Poincaré ball to encode hierarchical depth within a static, pre-built knowledge graph, using dual-space retrieval that fuses Euclidean and hyperbolic rankings. However, HyperbolicRAG relies on Euclidean encoders to produce the initial embeddings, leaving the fundamental geometric mismatch. Moreover, aggregating token embeddings into document representations poses a challenge that existing work in hyperbolic learning does not address (Yang et al., 2024). As we show in Proposition 4.3, naively averaging tokens in Euclidean space before projecting to hyperbolic space causes representations to collapse toward the origin, destroying the hierarchical structure that is meant be to preserved.

To this end, we introduce hyperbolic dense retrieval for RAG, framing embedding geometry as a design choice for improving evidence selection and grounding at the representation level. We study this through two complementary instantiations. First, HyTE-FH (Hyerbolic Text Encoder, Fully Hyperbolic) operates entirely in the Lorentz model of hyperbolic space, enabling end-to-end representation learning. Second, HyTE-H (Hybrid) maps embeddings from off-the-shelf Euclidean encoders into hyperbolic space, allowing us to build on existing pre-trained Euclidean models. The Lorentz model's intrinsic geometry enables parameter-efficient scaling: HyTE-H outperforms Euclidean baselines several times (2-3x) its size, reducing memory footprint in resource-constrained settings. To address the aggregation challenge in both instantiations, we introduce the Outward Einstein Midpoint, a geometry-aware pooling operator that amplifies tokens farther from the origin, provably preserving hierarchical structure during pooling.

Through extensive evaluation on RAGBench, we demonstrate that both hyperbolic variants consistently outperform Euclidean baselines in answer relevancy across multiple datasets, while achieving competitive performance on MTEB. Our experiments validate three key findings: (1) hyperbolic retrieval substantially improves RAG performance, with up to 29% gains over Euclidean baselines in context relevance and answer relevance; (2) hyperbolic models naturally encode concept-level hierarchies in their radial structure, with the fully hyperbolic model achieving a 20.2% radius increase from general to specific concepts, while Euclidean models fail to capture this organization; and (3) our theoretically grounded Outward Einstein Midpoint pooling preserves this hierarchical structure during aggregation.

## 2. Related Works

**Text Embeddings and Dense Retrieval.** Dense retrieval embeds queries and documents into a shared vector space and ranks candidates by similarity (*e.g.*, dot product or cosine). Transformer bi-encoders (*e.g.*, BERT (Devlin et al., 2019)) are widely used in this context due to their scalability with maximum inner product search (Karpukhin et al., 2020; Reimers & Gurevych, 2019). Most methods train with contrastive objectives using in-batch and hard negatives (Gao et al., 2021; Xiong et al., 2021), often following large-scale pretraining plus task-specific fine-tuning (Wang et al., 2022; Li et al., 2023; Nussbaum et al., 2025). More recently, decoder-only embedding models initialize from LLMs to exploit their pretrained linguistic knowledge (Muennighoff et al., 2024; Lee et al., 2024; Zhang et al., 2025). However, most retrievers remain reliant on inner products or distances in Euclidean geometry-an inductive bias often misaligned with the hierarchical structure of language and document collections. We address this gap by introducing hyperbolic geometry for text embeddings to better capture such a hierarchy.

**Retrieval Augmented Generation.** RAG grounds LLMs in retrieved evidence to improve factuality and access external knowledge (Oche et al., 2025). It typically retrieves top-$k$ contexts (often via dense retrieval) and conditions generation on them (Lewis et al., 2020). Since the context window is limited, retrieval quality is a key bottleneck for relevance and faithfulness (Friel et al., 2024a). Several methods improve reliability *after* retrieval: Self-RAG (Asai et al., 2024) and CRAG (Yan et al., 2024) use learned critics to filter or re-rank evidence, while GraphRAG (Han et al., 2024) leverages knowledge graphs for structured subgraph retrieval.

These approaches operate downstream of the embedding space and are complementary to ours geometric approach. Our goal is to improve RAG upstream by enhancing the retriever representations so that the initial top-$k$ evidence is more reliable under realistic efficiency constraints.

**Hyperbolic Representation Learning.** Hyperbolic geometry is primarily known for its ability to better capture hierarchical, tree-like structures (Yang et al., 2023; Peng et al., 2021), which enhances performance in various tasks, including molecular generation (Liu et al., 2019), recommendation (Yang et al., 2021; Li et al., 2021), image retrieval (Khrulkov et al., 2020; Wei et al., 2024; Bui et al., 2025), and knowledge graph embedding (Ganea et al., 2018a; Dhingra et al., 2018). More recently, hyperbolic geometry has also shown promise for multi-modal embedding models (Desai et al., 2023; Ibrahimi et al., 2024; Pal et al., 2024) and foundation models (Yang et al., 2025; He et al., 2025a). In contrast to these works, we study how hyperbolic representations can improve retrieval in RAG systems. Concurrently, Cao et al. (2025) use hyperbolic geometry to improve RAG rankings, but obtain hyperbolic embeddings via a simple projection from Euclidean encoders; by contrast, we build on fully hyperbolic encoders trained end-to-end and address key challenges in this setting, including providing the theoretically grounded geometry-aware pooling for document-level representations.

# 3. Hyperbolic Space Preliminaries

In this section, we go over all the preliminaries of Lorentz model of hyperbolic space and introduce the basic building blocks that create HyTE-FH.

## 3.1. Lorentz Model of Hyperbolic Space

We represent all embeddings in $d$-dimensional hyperbolic space $\mathbb{H}_K^d$ with constant negative curvature $K < 0$ using the Lorentz (hyperboloid) model. In the Lorentz model, hyperbolic space is realized as the upper sheet of a two-sheeted hyperboloid embedded in $\mathbb{R}^{d+1}$,

$$\mathbb{H}_K^d = \left\{ \mathbf{x} \in \mathbb{R}^{d+1} \,\middle|\, \langle \mathbf{x}, \mathbf{x} \rangle_L = \frac{1}{K}, \ x_0 > 0 \right\},$$

where the Lorentzian inner product is defined as $\langle \mathbf{x}, \mathbf{y} \rangle_L = -x_0 y_0 + \sum_{i=1}^d x_i y_i$. This formulation admits closed-form expressions for geodesic distances, barycentric operations, and parallel transport, and expresses similarity directly through Lorentzian inner products. The geodesic distance between two points $\mathbf{x}, \mathbf{y} \in \mathbb{H}_K^d$ is given by $d_K(\mathbf{x}, \mathbf{y}) = \frac{1}{\sqrt{-K}} \cosh^{-1}(K \langle \mathbf{x}, \mathbf{y} \rangle_L)$, which is a monotone function of the Lorentzian inner product.

To support optimization, we make use of exponential and logarithmic maps between the manifold and its tangent spaces. For a point $\mathbf{x} \in \mathbb{H}_K^d$, the logarithmic map $\log_x(\cdot)$ maps nearby points to the tangent space $T_x \mathbb{H}_K^d$, while the exponential map $\exp_x(\cdot)$ maps tangent vectors back to the manifold. These operators are used only where necessary for gradient-based updates, ensuring that all representations remain on $\mathbb{H}_K^d$ and preserving the hierarchical structure induced by negative curvature.

## 3.2. Hyperbolic Transformer Components

Standard operations cannot be applied directly in hyperbolic space, as they may violate the manifold constraint (Yang et al., 2024). To address this, we introduce hyperbolic components that serve as the building blocks for our embedding model. These operations are performed via a re-centering procedure that applies Euclidean operations in a latent space and maps the result back to the Lorentz model. By doing so, the resulting vector is constructed to satisfy the Lorentz constraint, thereby preserving the hyperbolic structure of representations. We present these operations as follows.

**Lorentz Linear Layer.** Given curvatures $K_1, K_2$, and parameters $\mathbf{W} \in \mathbb{R}^{(n+1) \times m}$ and $\mathbf{b} \in \mathbb{R}^m$ with $\mathbf{z} = |\mathbf{W}^\top \mathbf{x} + \mathbf{b}|$, the Lorentzian linear transformation (Yang et al., 2024) is the map $\mathrm{HLT} : \mathbb{L}^{K_1, n} \to \mathbb{L}^{K_2, m}$ given by,

$$\mathrm{HLT}(\mathbf{x}; \mathbf{W}, \mathbf{b}) = \sqrt{\frac{K_2}{K_1}} \cdot \left[ \sqrt{\|\mathbf{z}\|^2 - 1/K_2}, \mathbf{z} \right]$$

**Hyperbolic Layer Normalization.** Given token embeddings $X = \{\mathbf{x}_i\}_{i=1}^n \subset \mathbb{H}_K^d$, hyperbolic layer normalization is defined as

$$\mathrm{HypLayerNorm}(X) = \left( \sqrt{\frac{K_1}{K_2} \|\mathbf{z}\|_2^2 - \frac{1}{K_2}}, \sqrt{\frac{K_1}{K_2}} \mathbf{z} \right)$$

where $z = f_{\mathrm{LN}}(\mathbf{x}_{i,[1:d]})$, $f_{\mathrm{LN}}(\cdot)$ denotes standard Euclidean LayerNorm applied to the spatial components of the embedding, and $K_1, K_2 > 0$ are input and output curvature respectively.

**Lorentz Residual Connection.** Let $\mathbf{x}, f(\mathbf{x}) \in \mathbb{L}^{K,n}$ where $\mathbf{x}$ is an input vector and $f(\mathbf{x})$ is the output of a neural network $f$. Then, the Lorentzian residual connection (He et al., 2025d) is given by $\mathbf{x} \oplus_{\mathcal{L}} f(\mathbf{x}) = \alpha_1 \mathbf{x} + \alpha_2 \mathbf{y}$, where

$$\alpha_i = w_i / \left( \sqrt{-K} \|w_1 \mathbf{x} + w_2 f(\mathbf{x})\|_{\mathcal{L}} \right), \text{ for } i \in \{0, 1\},$$

where $\alpha_1, \alpha_2$ are weights parametrized by constants $(w_1, w_2) \in \mathbb{R}^2 \setminus \{(0,0)\}$.

**Hyperbolic Self-Attention.** In hyperbolic attention, similarity is governed by hyperbolic geodesic distance (Ganea et al., 2018b). Given token embeddings $X = \{\mathbf{x}_i\}_{i=1}^n \subset \mathbb{H}_K^d$, queries, keys, and values are computed via Lorentz-linear transformations as $\mathbf{Q} = \mathrm{HLT}(X; \mathbf{W}^Q, \mathbf{b}^Q)$, $\mathbf{K} =$

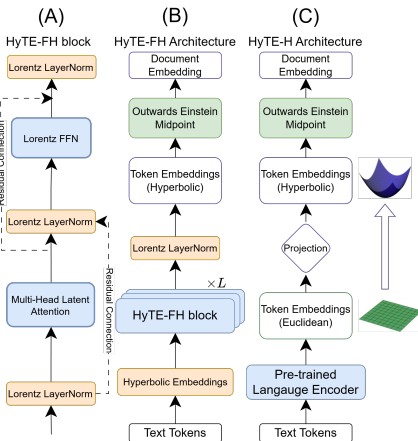

*Figure 2.* **HyTE Architecture**. A) HyTE-FH Encoder Block, B) HyTE-FH architecture, C) HyTE-H Architecture.

$\text{HLT}(X; \mathbf{W}^K, \mathbf{b}^K)$, and $\mathbf{V} = \text{HLT}(X; \mathbf{W}^V, \mathbf{b}^V)$, where $\text{HLT}(\cdot)$ denotes a linear map in Lorentz space. Attention weights are computed using squared hyperbolic geodesic distances (He et al., 2025c; Chen et al., 2022) as

$$\nu_{i,j} = \frac{\exp\left(-d_K^2(\mathbf{q}_i, \mathbf{k}_j)/\sqrt{m}\right)}{\sum_{l=1}^n \exp\left(-d_K^2(\mathbf{q}_i, \mathbf{k}_l)/\sqrt{m}\right)},$$

with head dimension $m$. This prioritizes geodesic proximity rather than angular similarity. The attended representation is obtained via a Lorentzian weighted midpoint

$$\text{Att}_{\mathcal{L}}(\mathbf{x})_i = \frac{\sum_{j=1}^n \nu_{i,j} \lambda_j \mathbf{v}_j}{\sqrt{-K} \left\| \sum_{j=1}^n \nu_{i,j} \lambda_j \mathbf{v}_j \right\|_{\mathcal{L}}},$$

where $\lambda_j = v_{j,0}$ is the Lorentz factor. Unlike Euclidean averaging, this aggregation remains on $\mathbb{H}_K^d$ and preserves radial structure during contextualization.

# 4. Method

We now outline our approach to hyperbolic dense retrieval. We begin by introducing the two proposed HyTE architectures, followed by an analysis of why naïve pooling strategies fail in hyperbolic space, and conclude by presenting our geometry-aware aggregation operator.

## 4.1. Architecture

The hyperbolic encoder components described in Section 3 form the building blocks (Figure 2A) of HyTE-FH, our fully hyperbolic transformer (Figure 2B). By operating entirely within hyperbolic geometry, HyTE-FH preserves hierarchical structure throughout token-level contextualization, aggregation, and similarity computation, with semantic abstraction and specificity encoded along radial dimensions. HyTE-H (Figure 2C) instead projects pretrained Euclidean representations into hyperbolic space, which allows hyperbolic geometry to be leveraged with a strong initialization

and avoiding the need to train a fully hyperbolic encoder from scratch.

While hyperbolic self-attention enables geometry-consistent contextualization at the token level, dense retrieval requires aggregating variable-length sequences into fixed-dimensional representations. Standard approaches map representations to tangent space, aggregate in Euclidean space, then map back to the manifold (Yang et al., 2024; Desai et al., 2023), but this distorts hierarchical structure encoded in radial depth in both the models. In the following subsections, we analyze this failure mode formally and introduce a pooling operator designed to preserve hierarchical information.

## 4.2. Failure of Naïve Hyperbolic Pooling

Naïve pooling strategies that aggregate in Euclidean space (Yang et al., 2024; Desai et al., 2023) systematically contract representations toward the origin. This follows from hyperbolic convexity: for any $\{\mathbf{x}_i\}_{i=0}^n \subset \mathbb{H}_K^d$, the barycenter lies strictly closer to the origin than the maximum-radius point unless all points coincide. Consequently, document-level embeddings lose the radial separation that encodes document specificity through hierarchical depth. To address this failure mode, we first establish notation for projecting ambient vectors onto the hyperboloid and measuring radial depth.

**Definition 4.1** (Lorentz Projection). For $\mathbf{v} \in \mathbb{R}^{d+1}$ with $\langle \mathbf{v}, \mathbf{v} \rangle_L < 0$ and $v_0 > 0$, let $\Pi_K(\mathbf{v}) = \frac{\mathbf{v}}{\sqrt{K \langle \mathbf{v}, \mathbf{v} \rangle_L}}$ denote the unique positive rescaling satisfying $\langle \Pi_K(\mathbf{v}), \Pi_K(\mathbf{v}) \rangle_L = 1/K$.

**Definition 4.2** (Radial Depth). The radial depth of $\boldsymbol{x} \in \mathbb{H}_K^d$ is $r(\mathbf{x}) = x_0$. Since $x_0 = \frac{1}{\sqrt{-K}} \cosh(\sqrt{-K}\, \rho)$ where $\rho = d_K(o, \mathbf{x})$, ordering by $x_0$ is equivalent to ordering by intrinsic hyperbolic distance from the origin.

Semantically, radial depth encodes *concept specificity*: general concepts should lie near the origin while fine-grained entities should have larger radii. This provides a measurable signature for evaluating whether models learn meaningful hierarchical structure. The simplest aggregation strategy is Euclidean averaging in the ambient space followed by reprojection. However, this approach provably contracts representations toward the origin (Ganea et al., 2018a; Chami et al., 2019), destroying hierarchical structure encoded in radial depth. We formalize this in the following proposition.

**Proposition 4.3** (Euclidean Mean Contracts). *Let* $\{\mathbf{x}_i\}_{i=1}^n \subset \mathbb{H}_K^d$ *with* $n \geq 2$. *Define the Euclidean mean* $\bar{\mathbf{x}} = \frac{1}{n}\sum_{i=1}^n \mathbf{x}_i$ *and its projection onto the hyperboloid* $\mathbf{m}^{\text{Euc}} = \Pi_K(\bar{\mathbf{x}})$. *Then, we have* $r(\mathbf{m}^{\text{Euc}}) \leq \frac{1}{n}\sum_{i=1}^n r(\mathbf{x}_i)$, *with equality if and only if all* $\mathbf{x}_i$ *are identical.*

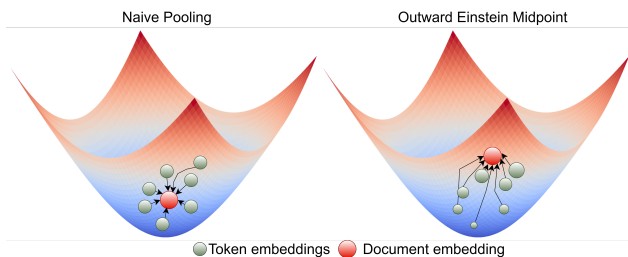

**Naive Pooling**     **Outward Einstein Midpoint**

○ Token embeddings   ● Document embedding

*Figure 3.* Outward Einstein Midpoint. Size of token shows its contribution towards aggregation.

The proof of this Proposition is available in Appendix A.2. This failure motivates a precise characterization of desirable pooling behavior. We formalize the requirement that pooling should preserve, rather than collapse, radial structure.

**Definition 4.4** (Outward Bias). *A pooling operator* $\mathcal{P}$ : $(\mathbb{H}_K^d)^n \to \mathbb{H}_K^d$ *is outward-biased if* $r(\mathcal{P}(\{\mathbf{x}_i\}_{i=1}^n)) \geq \bar{r}$, *where* $\bar{r}$ *is the weighted mean radius.*

A natural alternative is a weighted aggregation scheme in which token contributions are modulated by their relative importance. For example, Zhu et al. (2020) adopt the Einstein midpoint, the canonical barycenter in hyperbolic space (Gulcehre et al., 2019), to emphasize semantically specific tokens during pooling: since points near the boundary receive higher weight via the Lorentz factor $\lambda_i = x_{i,0}$, more informative content should dominate the aggregate. However, we show this intuition is misleading: the implicit radial weighting is fundamentally insufficient to counteract hyperbolic contraction at the document level.

**Proposition 4.5** (Implicit Radial Weighting is Insufficient). *The Einstein midpoint weights points by the Lorentz factor* $\lambda_i = x_{i,0}$, *but this weighting grows as* $\exp(\sqrt{-K}\rho)$ *while hyperbolic volume grows as* $\exp((d-1)\sqrt{-K}\rho)$. *Specifically, for a point* $\mathbf{x} \in \mathbb{H}_K^d$ *at hyperbolic distance* $\rho$ *from the origin* $o = (1/\sqrt{-K}, 0, \ldots, 0)$, *we have*

$$x_0 = \frac{1}{\sqrt{-K}} \cosh\left(\sqrt{-K}\,\rho\right) \sim \frac{1}{2\sqrt{-K}} \exp\left(\sqrt{-K}\,\rho\right)$$

*as* $\rho \to \infty$. *Thus, the Lorentz factor weighting undercompensates for the exponential growth of hyperbolic balls at large radii by a factor of* $\exp((d-2)\sqrt{-K}\,\rho)$.

These results establish that neither Euclidean averaging nor the standard Einstein midpoint satisfies the outward-bias property required for hierarchy-preserving aggregation. This motivates the design of a pooling operator with explicit radial amplification. The proof of this Proposition is available in Appendix A.3.

### 4.3. Outward Einstein Midpoint Pooling

To mitigate radial contraction during aggregation, we introduce the *Outward Einstein Midpoint*, a geometry-aware

pooling operator that explicitly amplifies the contribution of tokens with larger hyperbolic radius. Let $\{\mathbf{x}_i\}_{i=1}^n \subset \mathbb{H}_K^d$ denote a sequence of token embeddings, with optional attention weights $w_i \geq 0$, and $\lambda_i$ denoting the Lorentz factors. We define a radius-dependent weighting function

$$\phi_p(x_i) = x_{i,0}^p, \qquad p > 0,$$

which is monotone in the radial coordinate. The Outward Einstein Midpoint is then given by

$$\mathbf{m}_{K,p}^{\text{OEM}} = \frac{\sum_{i=1}^n \left(w_i\,\phi_p(\mathbf{x}_i)\right)\lambda_i\mathbf{x}_i}{\sum_{i=1}^n \left(w_i\,\phi_p(\mathbf{x}_i)\right)\lambda_i},$$

followed by reprojection onto the hyperboloid $\mathbb{H}_K^d$.

As shown in Figure 3, by construction, this operator assigns disproportionately higher weight to tokens located farther from the origin, counteracting the contraction inherent to naïve averaging. We now establish theoretical guarantees for the Outward Einstein Midpoint, showing that it systematically improves upon the standard Einstein midpoint in preserving radial structure.

**Theorem 4.6** (OEM Pre-Projection Bound). *Let* $\tilde{\mathbf{v}} = \sum_{i=1}^n \tilde{w}_i\mathbf{x}_i$ *where* $\tilde{w}_i \propto w_i x_{i,0}^{p+1}$ *are the normalized OEM weights. Then, for* $p \geq 0$, *we have*

$$\tilde{v}_0 = \frac{\sum_{i=1}^n w_i x_{i,0}^{p+2}}{\sum_{i=1}^n w_i x_{i,0}^{p+1}} \geq \frac{\sum_{i=1}^n w_i x_{i,0}}{\sum_{i=1}^n w_i} = \bar{r}_w.$$

We apply Chebyshev's sum inequality to the co-monotonic sequences $a_i = x_{i,0}^{p+1}$ and $b_i = x_{i,0}$ to prove this. Full proof can be found in Appendix A.4. While projection onto $\mathbb{H}_K^d$ contracts the radial coordinate, the OEM's concentration of weight on high-radius tokens inflates the pre-projection average, counteracting this effect. Theorem 4.6 establishes that OEM increases the pre-projection radial coordinate. The following theorem shows a stronger result: OEM provably dominates the standard Einstein midpoint in preserving radial structure.

**Theorem 4.7** (OEM Outward Bias). *Let* $\mathbf{m}_K^{\text{Ein}}$ *denote the standard Einstein midpoint* ($p = 0$) *and* $\mathbf{m}_{K,p}^{\text{OEM}}$ *the Outward Einstein Midpoint. Then, for all* $p \geq 1, r(\mathbf{m}_{K,p}^{\text{OEM}}) \geq r(\mathbf{m}_K^{\text{Ein}})$.

The OEM weights $\tilde{w}_i \propto w_i x_{i,0}^{p+1}$ concentrate more mass on high-radius points than the Einstein weights $w_i x_{i,0}$, increasing the pre-projection time component while reducing pairwise dispersion. Full proof in Appendix A.5. Together, these results establish that the Outward Einstein Midpoint provably preserves hierarchical structure during aggregation, in contrast to both Euclidean averaging and the standard Einstein midpoint. We validate this empirically through concept-level hierarchy analysis (Section 5.2), showing that

models using OEM pooling maintain monotonically increasing radii across semantic specificity levels—a property absent in Euclidean baselines.

## 4.4. Training Methodology

Following (Nussbaum et al., 2025), we train the hyperbolic encoder in three stages, with all objectives operating directly on the Lorentz manifold using geodesic-based similarity.

**Stage 1: Hyperbolic Masked Language Modeling.** We initialize via masked language modeling (MLM), following the standard BERT objective in hyperbolic space. Contextualization is performed through hyperbolic self-attention, with all intermediate representations on the hyperboloid. Predictions are produced using a Lorentzian multinomial logistic regression (LorentzMLR) (Bdeir et al., 2024) head, which defines class logits via Lorentzian inner products. Only HyTE-FH is trained on MLM, while for HyTE-H we choose a pre-trained Euclidean model as the MLM base to leverage a sronger initialization in low-resource settings.

**Stage 2: Contrastive Pre-Training.** We initialize the embedding model from the MLM model by adding the OEM pooler, and then train it on a large collection of general-domain query–document pairs. This stage encourages the model to learn representations that distinguish relevant documents from irrelevant ones. We use the standard InfoNCE loss (Oord et al., 2018), with similarity defined as the negative geodesic distance $s(q, d) = -d_K(q, d)$. For a batch $\mathcal{B}$, the contrastive loss is

$$\mathcal{L}_{\mathrm{ctr}} = -\sum_{i \in \mathcal{B}} \log \frac{\exp(s(\mathbf{q}_i, \mathbf{d}_i)/\tau)}{\sum_{j \in \mathcal{B}} \exp(s(\mathbf{q}_i, \mathbf{d}_j)/\tau)},$$

where $\tau > 0$ is a temperature and $B$ is the batch size. Following Nussbaum & Duderstadt (2025), we use a large batch size of 16384 at this stage.

**Stage 3: Contrastive Learning Fine-tuning.** In the final stage of training, we further fine-tune the embedding model using curated query–document data. For each training sample $i$, we augment the training data with hard negatives $\mathcal{H}_i$ mined following Nussbaum & Duderstadt (2025). The contrastive loss function becomes

$$\mathcal{L}_{\mathrm{ctr}} = -\sum_{i \in \mathcal{B}} \log \frac{\exp(s(\mathbf{q}_i, \mathbf{d}_i)/\tau)}{\sum_{j \in \mathcal{B} \cup \mathcal{H}_i} \exp(s(\mathbf{q}_i, \mathbf{d}_j)/\tau)}.$$

This stage refines retrieval behavior beyond unsupervised co-occurrence structure.

**Retrieval-Augmented Generation.** At inference time, the trained hyperbolic encoder is used to retrieve the top-$k$ documents $\mathcal{C}$ for a given query. These retrieved documents are then provided as context to a downstream generative language model. Prompt formatting and generation follow standard practice and are provided in Appendix B.

*Table 1.* Performance on MTEB benchmark. We report mean scores across tasks and task types. HyTE-FH performs best among the three models.

| Model | Mean (Task) | Mean (TaskType) |
|---|---|---|
| EucBERT | 54.11 | 51.31 |
| HyTE-H$^{\mathrm{Euc}}$ | 54.57 | 53.71 |
| HyTE-FH | **56.41** | **53.75** |

**Approximate Nearest Neighbor Search.** Practical dense retrieval requires approximate nearest neighbor (ANN) indexing, raising a natural concern: does standard ANN infrastructure, built around Euclidean inner products, apply to hyperbolic embeddings? It does, with no custom distance functions or index modifications. Hyperbolic nearest-neighbor search in the Lorentz model reduces cleanly to maximum inner product search (MIPS) over a simple transformation of the embeddings. The geodesic distance (Section 3), $d_K(x, y) = \frac{1}{\sqrt{-K}} \operatorname{arccosh}(K \langle x, y \rangle_L)$, is monotone in the Lorentzian inner product $\langle x, y \rangle_L$, so ranking by hyperbolic proximity is equivalent to ranking by $\langle x, y \rangle_L$. Writing $x = (x_0, \mathbf{x})$ and $y = (y_0, \mathbf{y})$ with the time coordinate first gives $\langle x, y \rangle_L = -x_0 y_0 + \mathbf{x}^\top \mathbf{y}$; negating each document's time coordinate ($x_0 \to -x_0$) turns this into the standard dot product $x_0 y_0 + \mathbf{x}^\top \mathbf{y}$, reducing hyperbolic ranking to MIPS on the sign-flipped embeddings. Exact retrieval thus runs through FAISS `IndexFlatIP` unmodified, and graph-based methods such as HNSW work out of the box: HNSW builds its proximity graph from pairwise comparisons alone, and each comparison on the sign-flipped embeddings correctly evaluates the Lorentzian inner product, so the graph topology faithfully reflects hyperbolic neighborhoods. HyTE is therefore a drop-in replacement for Euclidean encoders in any existing retrieval stack. Runtime and computational complexity appear in Appendix D.

## 5. Experiments and Results

### 5.1. Experimental Setup

**Datasets.** We pre-train our models using publicly available corpora following the data curation and filtering protocols introduced in nomic-embed (Nussbaum et al., 2025). For masked language modeling (MLM), we use the high-quality 2023 Wikipedia dump, which provides broad topical coverage and long-form text suitable for learning general-purpose semantic representations. For contrastive pre-training, we leverage approximately 235 million text pairs curated and filtered as described in (Nussbaum et al., 2025), designed to encourage semantic alignment across paraphrases and related content at scale. Finally, for task-specific fine-tuning, we use the training splits of the BEIR benchmark (Thakur et al., 2021), which comprises a diverse collection of retrieval tasks spanning multiple domains and query styles.

**Evaluation Benchmarks.** We evaluate our approach on

*Table 2.* RAG benchmark results comparing our model variants.

| Model | Average | | | CovidQA | | | Cuad | | | Emanual | | | DelucionQA | | | ExpertQA | | |
|---|---|---|---|---|---|---|---|---|---|---|---|---|---|---|---|---|---|---|
| | F | CR | AR | F | CR | AR | F | CR | AR | F | CR | AR | F | CR | AR | F | CR | AR |
| EucBERT | 0.596 | 0.798 | 0.647 | 0.685 | 0.863 | 0.582 | 0.654 | 0.644 | 0.641 | 0.642 | 0.646 | 0.674 | 0.525 | **0.968** | 0.679 | 0.475 | 0.872 | 0.662 |
| HyTE-H$^{\text{Euc}}$ | 0.706 | 0.814 | 0.739 | 0.708 | 0.868 | 0.668 | **0.787** | 0.652 | 0.710 | **0.679** | **0.835** | **0.814** | 0.737 | 0.857 | 0.773 | 0.623 | 0.859 | 0.728 |
| HyTE-FH | **0.732** | **0.848** | **0.765** | **0.764** | **0.916** | **0.694** | 0.747 | **0.674** | **0.752** | 0.660 | 0.807 | 0.704 | **0.789** | 0.906 | **0.861** | **0.702** | **0.936** | **0.814** |

F = Faithfulness, CR = Context Relevance, AR = Answer Relevance. Best results in bold.

*Table 3.* RAG benchmark results comparing our hybrid model with state-of-the-art embedding models. HyTE-H demonstrates competitive performance particularly in context relevance and answer relevance.

| Model | Average | | | CovidQA | | | Cuad | | | Emanual | | | DelucionQA | | | ExpertQA | | |
|---|---|---|---|---|---|---|---|---|---|---|---|---|---|---|---|---|---|---|
| | F | CR | AR | F | CR | AR | F | CR | AR | F | CR | AR | F | CR | AR | F | CR | AR |
| ModernBert* | 0.617 | 0.748 | 0.632 | 0.656 | 0.895 | 0.537 | 0.632 | 0.709 | 0.746 | 0.567 | 0.715 | 0.639 | 0.655 | 0.665 | 0.518 | 0.575 | 0.758 | 0.718 |
| GTE | 0.659 | 0.701 | 0.650 | 0.695 | 0.840 | 0.538 | 0.733 | 0.599 | 0.779 | 0.546 | 0.608 | 0.686 | 0.648 | 0.725 | 0.549 | 0.672 | 0.731 | 0.698 |
| Gemma | 0.603 | 0.735 | 0.684 | 0.685 | 0.760 | 0.497 | 0.724 | 0.600 | 0.778 | 0.555 | 0.884 | 0.687 | 0.612 | 0.643 | 0.705 | 0.442 | 0.791 | 0.755 |
| KaLM-mini-v1 | 0.624 | 0.719 | 0.591 | 0.656 | 0.787 | 0.528 | 0.742 | 0.789 | 0.716 | 0.565 | 0.776 | 0.616 | 0.553 | 0.581 | 0.573 | 0.607 | 0.666 | 0.522 |
| Qwen-Embedding | 0.784 | 0.907 | 0.814 | 0.808 | 0.955 | 0.747 | 0.825 | 0.738 | 0.806 | **0.838** | **0.973** | **0.852** | 0.701 | 0.881 | 0.840 | 0.748 | **0.988** | 0.824 |
| HyTE-H$^{\text{bert}}$ | 0.763 | 0.904 | 0.832 | 0.797 | 0.974 | **0.755** | 0.760 | 0.683 | 0.804 | 0.688 | 0.943 | **0.899** | **0.829** | 0.965 | **0.871** | 0.739 | 0.958 | 0.834 |
| HyTE-H$^{\text{Qwen}}$ | **0.803** | **0.934** | **0.845** | **0.832** | **1.000** | 0.747 | **0.886** | **0.855** | **0.960** | 0.826 | 0.963 | 0.764 | 0.809 | **0.987** | 0.822 | 0.666 | 0.863 | **0.931** |

F = Faithfulness, CR = Context Relevance, AR = Answer Relevance. Best results in bold.

two complementary benchmarks: (1) the Massive Text Embedding Benchmark (MTEB) (Muennighoff et al., 2023) to assess embedding quality across diverse tasks, and (2) RAGBench (Friel et al., 2024b) for end-to-end RAG system evaluation. In MTEB, we particularly use the English part of the benchmark. RAGBench evaluates RAG systems on domain-specific question-answering datasets including CovidQA, Cuad, Emanual, DelucionQA, and ExpertQA.

**Baselines.** We adopt different baseline strategies for our two models based on their training paradigms. For HyTE-FH, which is pre-trained from scratch, we train a fully Euclidean equivalent called EucBERT using the same architecture and training setup. This controlled comparison isolates the contribution of hyperbolic geometry. We also evaluate HyTE-H$^{\text{Euc}}$, a hybrid hyperbolic model initialized with EucBERT. The three models are evaluated on MTEB and RAGBench. For HyTE-H$^{\text{bert}}$, which is fine-tuned with modernbert-base (Warner et al., 2024) as base model, we compare against state-of-the-art embedding models smaller than 500M parameters, including gte-multilingual-base (Zhang et al., 2024), KaLM-embedding-multilingual-mini-v1 (Hu et al., 2025), and embeddinggemma-300m (Vera et al., 2025).

**Metrics.** For MTEB, we report mean scores across tasks and task types. For RAG evaluation, we measure three key metrics using RAGAS (Es et al., 2024): (1) *Faithfulness*, which assesses whether generated answers are grounded in the retrieved context; (2) *Context Relevance*, which measures how relevant the retrieved documents are to the query; and (3) *Answer Relevance*, which evaluates how well the generated answer addresses the user's question.

**Implementation.** We implement all hyperbolic models

using HyperCore (He et al., 2025e) and train on NVIDIA H100 GPUs. All three models, HyTE-FH, HyTE-H, and EucBERT, share the same architecture, each containing 149M parameters with 12 transformer layers and 768-dimensional embeddings. For generation and judging, we use Llama-3.1-8B-Instruct (Weerawardhena et al., 2025). For RAG benchmarks, we fix the retrieval context window size to 5 for all models to ensure a controlled comparison; we additionally report ablations with larger context sizes in Appendix Table A4.

### 5.2. Results

**MTEB Benchmark.** Table 1 reports performance on the MTEB benchmark. HyTE-FH achieves the highest mean score across tasks (56.41), outperforming both EucBERT (54.11) and HyTE-H$^{\text{Euc}}$ (54.57). On the task-type mean, HyTE-FH and HyTE-H$^{\text{Euc}}$ perform comparably (53.75 and 53.71, respectively), with both surpassing EucBERT (51.31). These results demonstrate that hyperbolic representations not only improve RAG retrieval but also remain competitive on general-purpose embedding benchmarks. We present task-wise results in Table A2.

**RAG Benchmark Results.** Table 2 presents RAG benchmark results across five datasets. HyTE-FH achieves the best average performance across all three metrics: faithfulness (0.732), context relevance (0.848), and answer relevance (0.765). HyTE-H$^{\text{Euc}}$ ranks second overall, with both hyperbolic variants substantially outperforming EucBERT. On individual datasets, HyTE-FH leads on CovidQA, Cuad, DelucionQA, and ExpertQA, while HyTE-H$^{\text{Euc}}$ achieves the best context and answer relevance on Emanual. These results demonstrate that hyperbolic geometry consistently

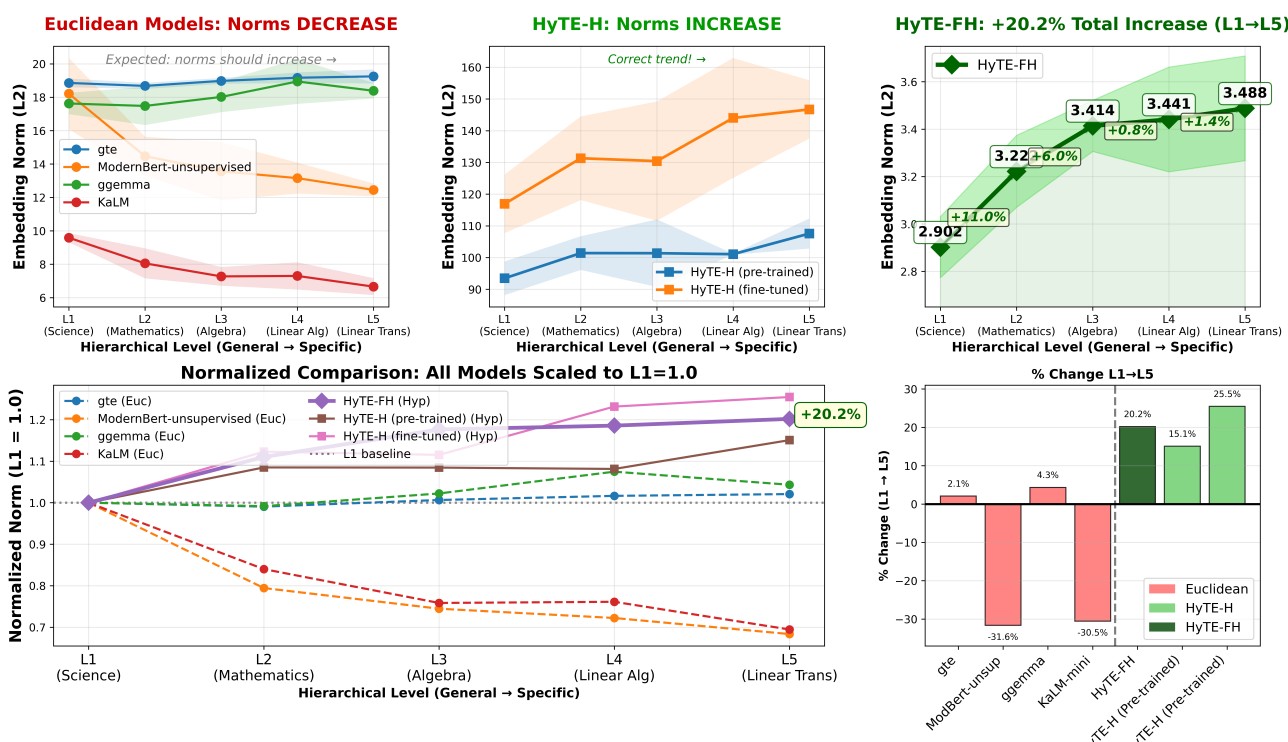

**Figure 4. Empirical validation of hierarchical encoding. Left:** Euclidean models show flat or decreasing norms. **Middle:** HyTE-H demonstrate increasing norms with fine-tuning enhancing this trend. **Right:** HyTE-FH achieves +20.2% total increase from L1 to L5. **Bottom:** Normalized comparison and percent change summary highlighting the contrasting behaviors of different geometric approaches.

improves retrieval quality for RAG across diverse domains.

Table 3 reports RAG performance across five datasets. HyTE-H[bert] consistently outperforms strong Euclidean embedding baselines across all metrics, with particularly large gains in context relevance and answer relevance. These improvements indicate that hyperbolic representations are more effective at retrieving structurally relevant evidence, which is critical for downstream generation quality in RAG pipelines. In qualitative case studies shows in Appendix F.1, we observe that Euclidean models frequently fail to retrieve key supporting passages altogether, whereas hyperbolic model recover relevant evidence more reliably, leading to more faithful and contextually grounded answers.

**Concept-Level Hierarchy Analysis.** A central motivation for hyperbolic embeddings is their capacity to preserve hierarchical relationships (Section 4.2). To understand how models capture document hierarchy, we analyze learned radii (distances from the origin in the Poincaré ball) across five hierarchical levels: from Level 1 (most general, e.g., document-level topics) to Level 5 (most specific, e.g., fine-grained entities). Figure 4 presents these results. The fully hyperbolic model demonstrates clear hierarchical organization with radii increasing monotonically from Level 1 (2.902) to Level 5 (3.488, +20.2%). This shows the model

naturally places general concepts near the origin and specific details toward the boundary, consistent with hyperbolic geometry, where proximity to the origin represents generality. Euclidean models show flat or decreasing distributions. Baselines maintain constant norms across levels or decreases norm by 30%, reflecting inverted structure. Hybrid models exhibit substantially larger radii from the hyperbolic component. The fine-tuned hybrid increases from 116.9 to 146.7, showing that fine-tuning induces structured hierarchy. We have attached the dataset for this case study in the supplementary material. The concept level hierarchy data is available in Appendix C.

**ANN Retrieval on MS MARCO.** To confirm the MIPS reduction yields practical gains, we index HyTE-FH and EucBERT on MS MARCO (100k passages, 6,980 dev queries) with FAISS IndexHNSWFlat, applying the sign-flip conversion to HyTE-FH. Table 4 reports Recall@k against the exact top-k ground truth and end-to-end latency. HyTE-FH beats EucBERT at every k (by 6.4 points at k=50 and 4.3 at k=100) while running 3x faster (0.09 vs. 0.29 ms); build times are identical ( 10 s). We attribute this to OEM pooling: its outward bias produces well-separated embeddings with greater radial spread, giving the HNSW graph a more tree-like structure with fewer spurious shortcuts, so greedy traversal converges faster and more accurately. ANN search

on hyperbolic embeddings is thus not merely viable but more efficient than its Euclidean counterpart.

*Table 4.* ANN retrieval on MS MARCO (100k passages, 6,980 queries) using FAISS `IndexHNSWFlat`. HyTE-FH applies the sign-flip reduction to reuse the same Euclidean IP index. HyTE-FH achieves higher recall at every $k$ and $3\times$ lower latency.

| $k$ | HyTE-FH (Recall@$k$) | EucBERT (Recall@$k$) |
|---|---|---|
| 1 | **0.917** | 0.913 |
| 10 | **0.906** | 0.891 |
| 50 | **0.859** | 0.795 |
| 100 | **0.750** | 0.707 |

**Hierarchy-Sensitive Retrieval.** To directly test whether hyperbolic geometry helps distinguish documents at different levels of specificity, we construct a controlled diagnostic benchmark from the MeSH (Medical Subject Headings) taxonomy using PubMed abstracts, where each query must be matched to an abstract at a specific hierarchical depth against distractors drawn from its parent, grandparent, sibling, and unrelated branches (construction details in Appendix E). We evaluate in a zero-shot setting (no training on MeSH data) and report Recall@1 alongside Specificity Hit Rate (SpecHR), the fraction of queries where the correct-depth target is ranked above all ancestor-level documents. As shown in Table 5, HyTE-FH ranks the correct-depth document first 96.3% of the time versus 81.0% for EucBERT, with SpecHR improving from 0.883 to 0.970. The error distribution in Table 6 reveals why: EucBERT's failures concentrate at parent-level (7.0%) and sibling-level (7.0%) confusions, exactly the hierarchical failure modes our method is designed to address. HyTE-FH eliminates grandparent confusion entirely, confirming that hyperbolic geometry provides a concrete advantage precisely where it matters: distinguishing documents at different levels of specificity within a known hierarchy.

*Table 5.* Zero-shot results on the MeSH hierarchy-sensitive retrieval benchmark. SpecHR is the fraction of queries where the correct-depth target is ranked above all ancestor-level documents.

| Model | SpecHR | R@1 | R@5 | MRR |
|---|---|---|---|---|
| EucBERT | 0.883 | 0.810 | 0.963 | 0.878 |
| HyTE-FH | **0.970** | **0.963** | **0.990** | **0.976** |

*Table 6.* Top-1 prediction breakdown on the MeSH benchmark by distractor type. HyTE-FH substantially reduces both parent and sibling confusions and eliminates grandparent errors.

| Model | Target | Sibling | Parent | Grandparent | Random |
|---|---|---|---|---|---|
| EucBERT | 81.0% | 7.0% | 7.0% | 1.0% | 4.0% |
| HyTE-FH | **96.3%** | 1.7% | 1.7% | 0.0% | 0.3% |

*Table 7.* Comparison of pooling strategies on MTEB tasks. OEM pooling leverages hyperbolic geometry for improved performance.

| Pooling Strategy | Mean (Task) | Mean (TaskType) |
|---|---|---|
| Naive Pooling | 47.07 | 45.61 |
| CLS Token | 49.33 | 48.90 |
| Einstein Midpoint | 50.33 | 47.19 |
| OEM | **56.41** | **53.75** |

**Ablation Studies.** We compare two pooling strategies for aggregating token embeddings into document representations: CLS token pooling and OEM pooling. CLS pooling uses the representation of a special classification token, while OEM pooling performs geometry-aware aggregation directly in hyperbolic space. Table 7 shows that OEM outperforms all alternatives by a substantial margin, improving over the next-best method (Einstein Midpoint) with a 10% improvement. Naive mean pooling performs worst, consistent with the radial collapse predicted by Proposition 4.3, while the standard Einstein midpoint improves over CLS but remains well below OEM, confirming that the outward bias is essential for preserving hierarchical structure.

We also show that using geodesic distance in the contrastive objective outperforms the Lorentz inner product (Appendix Table A3), suggesting better alignment of representations on the manifold. Additionally, hyperbolic models maintain strong performance with smaller retrieval budgets, whereas Euclidean baselines require larger context windows to achieve comparable results (Appendix Table A4). A sensitivity analysis over $p$ (Table A5) shows that performance peaks at $p = 1.0$ and degrades gracefully on either side, with $p = 0$ recovering the standard Einstein midpoint and larger $p$ over-amplifying peripheral tokens. We therefore use $p = 1.0$ throughout. In Table A6, we also show that the hyperbolic model mitigates the problem of hubness and has lower number of hubs compared to Euclidean models.

# 6. Conclusion

We introduced hyperbolic dense retrieval for RAG, showing that aligning embedding geometry with the hierarchical structure of language improves faithfulness and answer quality. Our approach preserves document-level structure during aggregation through a geometry-aware pooling operator, addressing a key failure mode of Euclidean retrieval pipelines. Across evaluations, we observe consistent gains using models substantially smaller than current state-of-the-art retrievers, highlighting the effectiveness of hyperbolic inductive bias over scale alone. Case studies further show that hyperbolic representations organize documents by specificity through norm-based separation, a property absent in Euclidean embeddings. These findings suggest that embedding geometry is a central design choice for reliable retrieval in RAG systems.

## Impact Statement

This paper presents work whose goal is to advance the field of Machine Learning, specifically dense retrieval for retrieval-augmented generation systems. By improving the geometric fidelity of document embeddings, our approach aims to reduce retrieval errors that can lead to hallucinated or poorly grounded responses in RAG systems. We believe more accurate retrieval contributes positively to the reliability of AI-generated content. Additionally, our fully hyperbolic model demonstrates improved parameter efficiency, which may reduce computational costs and environmental impact associated with training and deploying embedding models. There are many potential societal consequences of our work, none which we feel must be specifically highlighted here.

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

# A. Proofs

Throughout, we work in the Lorentz model with curvature $K < 0$, where

$$\mathbb{H}_K^d = \{\mathbf{x} \in \mathbb{R}^{d+1} : \langle \mathbf{x}, \mathbf{x} \rangle_L = 1/K, \ x_0 > 0\}$$

and $\langle \mathbf{x}, \mathbf{y} \rangle_L = -x_0 y_0 + \sum_{i=1}^d x_i y_i$ denotes the Lorentzian inner product.

## A.1. Auxiliary Lemma

**Lemma A.1** (Lorentzian Inner Product Bound). *For any $\mathbf{x}, \mathbf{y} \in \mathbb{H}_K^d$, we have $K\langle \mathbf{x}, \mathbf{y} \rangle_L \geq 1$, with equality if and only if $\mathbf{x} = \mathbf{y}$.*

*Proof.* The geodesic distance on $\mathbb{H}_K^d$ satisfies

$$d_K(\mathbf{x}, \mathbf{y}) = \frac{1}{\sqrt{-K}} \cosh^{-1}(K\langle \mathbf{x}, \mathbf{y} \rangle_L).$$

Since $\cosh^{-1} : [1, \infty) \to [0, \infty)$ and $d_K(\mathbf{x}, \mathbf{y}) \geq 0$ with equality if and only if $\mathbf{x} = \mathbf{y}$, we conclude $K\langle \mathbf{x}, \mathbf{y} \rangle_L \geq 1$ with equality if and only if $\mathbf{x} = \mathbf{y}$. $\qquad\square$

## A.2. Proof of Proposition 4.3

**Proposition A.2** (Euclidean Mean Contracts). *Let $\{\mathbf{x}_i\}_{i=1}^n \subset \mathbb{H}_K^d$ with $n \geq 2$. Define the Euclidean mean $\bar{\mathbf{x}} = \frac{1}{n} \sum_{i=1}^n \mathbf{x}_i$ and its projection onto the hyperboloid $\mathbf{m}^{\mathrm{Euc}} = \Pi_K(\bar{\mathbf{x}})$. Then:*

$$r(\mathbf{m}^{\mathrm{Euc}}) \leq \frac{1}{n} \sum_{i=1}^n r(\mathbf{x}_i)$$

*with equality if and only if all $\mathbf{x}_i$ are identical.*

*Proof.* We first verify that the projection is well-defined, then establish the contraction inequality.

We must show $\langle \bar{\mathbf{x}}, \bar{\mathbf{x}} \rangle_L < 0$ and $\bar{x}_0 > 0$. The latter is immediate since $\bar{x}_0 = \frac{1}{n} \sum_i x_{i,0} > 0$. For the former, compute:

$$K\langle \bar{\mathbf{x}}, \bar{\mathbf{x}} \rangle_L = K \left\langle \frac{1}{n} \sum_i \mathbf{x}_i, \frac{1}{n} \sum_j \mathbf{x}_j \right\rangle_L = \frac{1}{n^2} \sum_{i,j} K\langle \mathbf{x}_i, \mathbf{x}_j \rangle_L.$$

By Lemma A.1, each term satisfies $K\langle \mathbf{x}_i, \mathbf{x}_j \rangle_L \geq 1$. Therefore:

$$K\langle \bar{\mathbf{x}}, \bar{\mathbf{x}} \rangle_L \geq \frac{1}{n^2} \cdot n^2 = 1 > 0.$$

Since $K < 0$, this implies $\langle \bar{\mathbf{x}}, \bar{\mathbf{x}} \rangle_L < 0$, confirming projectability.

The projection is given by $\mathbf{m}^{\mathrm{Euc}} = \bar{\mathbf{x}}/\sqrt{K\langle \bar{\mathbf{x}}, \bar{\mathbf{x}} \rangle_L}$, so the radial depth satisfies:

$$r(\mathbf{m}^{\mathrm{Euc}}) = \frac{\bar{x}_0}{\sqrt{K\langle \bar{\mathbf{x}}, \bar{\mathbf{x}} \rangle_L}}.$$

From Step 1, we have $K\langle \bar{\mathbf{x}}, \bar{\mathbf{x}} \rangle_L \geq 1$, hence $\sqrt{K\langle \bar{\mathbf{x}}, \bar{\mathbf{x}} \rangle_L} \geq 1$. Therefore:

$$r(\mathbf{m}^{\mathrm{Euc}}) = \frac{\bar{x}_0}{\sqrt{K\langle \bar{\mathbf{x}}, \bar{\mathbf{x}} \rangle_L}} \leq \bar{x}_0 = \frac{1}{n} \sum_{i=1}^n x_{i,0} = \frac{1}{n} \sum_{i=1}^n r(\mathbf{x}_i).$$

Equality holds if and only if $\sqrt{K\langle \bar{\mathbf{x}}, \bar{\mathbf{x}} \rangle_L} = 1$, which by Step 1 requires $K\langle \mathbf{x}_i, \mathbf{x}_j \rangle_L = 1$ for all pairs $i, j$. By Lemma A.1, this occurs if and only if $\mathbf{x}_i = \mathbf{x}_j$ for all $i, j$, i.e., all points are identical. $\qquad\square$

## A.3. Proof of Proposition 4.5

**Proposition A.3** (Implicit Radial Weighting is Insufficient). *The Einstein midpoint weights points by the Lorentz factor* $\lambda_i = x_{i,0}$, *but this weighting grows as* $\exp(\sqrt{-K}\,\rho)$ *while hyperbolic volume grows as* $\exp((d-1)\sqrt{-K}\,\rho)$. *The Lorentz factor weighting therefore undercompensates by a factor of* $\exp((d-2)\sqrt{-K}\,\rho)$ *for* $d \geq 3$.

*Proof.* We establish the asymptotic growth rates of the Lorentz factor and hyperbolic volume separately, then compare them.

**Step 1: Lorentz factor asymptotics.** The hyperbolic distance from the origin $\mathbf{o} = (1/\sqrt{-K}, 0, \ldots, 0)$ to a point $\mathbf{x} \in \mathbb{H}_K^d$ is:

$$\rho = d_K(\mathbf{o}, \mathbf{x}) = \frac{1}{\sqrt{-K}} \cosh^{-1}(K\langle \mathbf{o}, \mathbf{x}\rangle_L).$$

Computing the inner product:

$$\langle \mathbf{o}, \mathbf{x}\rangle_L = -\frac{x_0}{\sqrt{-K}},$$

so $K\langle \mathbf{o}, \mathbf{x}\rangle_L = -K \cdot (-x_0/\sqrt{-K}) = \sqrt{-K}\,x_0$. Thus:

$$\rho = \frac{1}{\sqrt{-K}} \cosh^{-1}(\sqrt{-K}\,x_0).$$

Inverting this relation:

$$x_0 = \frac{1}{\sqrt{-K}} \cosh(\sqrt{-K}\,\rho).$$

For large $\rho$, using $\cosh(t) \sim \frac{1}{2}e^t$:

$$x_0 \sim \frac{1}{2\sqrt{-K}} \exp(\sqrt{-K}\,\rho).$$

Hence the Lorentz factor $\lambda = x_0$ grows as $\exp(\sqrt{-K}\,\rho)$.

**Step 2: Hyperbolic volume asymptotics.** The volume of a geodesic ball of radius $\rho$ in $\mathbb{H}_K^d$ is:

$$\mathrm{Vol}(B_\rho) = \frac{\omega_{d-1}}{(-K)^{(d-1)/2}} \int_0^\rho \sinh^{d-1}(\sqrt{-K}\,t)\,dt,$$

where $\omega_{d-1} = 2\pi^{d/2}/\Gamma(d/2)$ is the surface area of the unit $(d-1)$-sphere. For large $\rho$, using $\sinh(t) \sim \frac{1}{2}e^t$:

$$\mathrm{Vol}(B_\rho) \sim C_d \exp((d-1)\sqrt{-K}\,\rho),$$

where $C_d$ is a dimension-dependent constant.

**Step 3: Compensation deficit.** The ratio of volume growth to Lorentz factor growth is:

$$\frac{\mathrm{Vol}(B_\rho)}{\lambda} \sim \frac{\exp((d-1)\sqrt{-K}\,\rho)}{\exp(\sqrt{-K}\,\rho)} = \exp((d-2)\sqrt{-K}\,\rho).$$

For $d \geq 3$, this ratio diverges as $\rho \to \infty$, demonstrating that the Lorentz factor provides insufficient compensation for the exponential growth of hyperbolic space at large radii. $\square$

## A.4. Proof of Theorem 4.6

**Theorem A.4** (OEM Pre-Projection Bound). *Let* $\tilde{\mathbf{v}} = \sum_{i=1}^n \tilde{w}_i \mathbf{x}_i$ *where* $\tilde{w}_i \propto w_i x_{i,0}^{p+1}$ *are the normalized OEM weights. Then, for* $p \geq 0$:

$$\tilde{v}_0 = \frac{\sum_{i=1}^n w_i \mathbf{x}_{i,0}^{p+2}}{\sum_{i=1}^n w_i \mathbf{x}_{i,0}^{p+1}} \geq \frac{\sum_{i=1}^n w_i \mathbf{x}_{i,0}}{\sum_{i=1}^n w_i} = \bar{r}_w.$$

*Proof.* We apply Chebyshev's sum inequality. Define sequences:

$$a_i = x_{i,0}^{p+1}, \qquad b_i = x_{i,0}.$$

Since $x_{i,0} > 0$ for all $i$ (points lie on the upper sheet of the hyperboloid) and $p \geq 0$, both sequences are strictly positive. Moreover, the sequences are co-monotonic: for any $i, j$,

$$x_{i,0} \geq x_{j,0} \iff x_{i,0}^{p+1} \geq x_{j,0}^{p+1},$$

since $t \mapsto t^{p+1}$ is strictly increasing on $(0, \infty)$.

Chebyshev's sum inequality states that for co-monotonic sequences $\{a_i\}$, $\{b_i\}$ and non-negative weights $\{w_i\}$ with $\sum_i w_i > 0$:

$$\left( \sum_i w_i a_i b_i \right) \left( \sum_i w_i \right) \geq \left( \sum_i w_i a_i \right) \left( \sum_i w_i b_i \right).$$

Substituting $a_i = x_{i,0}^{p+1}$ and $b_i = x_{i,0}$:

$$\left( \sum_i w_i x_{i,0}^{p+2} \right) \left( \sum_i w_i \right) \geq \left( \sum_i w_i x_{i,0}^{p+1} \right) \left( \sum_i w_i x_{i,0} \right).$$

Dividing both sides by $\left( \sum_i w_i x_{i,0}^{p+1} \right) \left( \sum_i w_i \right) > 0$:

$$\frac{\sum_i w_i x_{i,0}^{p+2}}{\sum_i w_i x_{i,0}^{p+1}} \geq \frac{\sum_i w_i x_{i,0}}{\sum_i w_i}.$$

The left-hand side equals $\tilde{v}_0$ and the right-hand side equals $\bar{r}_w$, completing the proof. Equality holds if and only if all $x_{i,0}$ are identical. $\square$

## A.5. Proof of Theorem 4.7

**Theorem A.5** (OEM Outward Bias). *Let $\mathbf{m}_K^{\mathrm{Ein}}$ denote the standard Einstein midpoint ($p = 0$) and $\mathbf{m}_{K,p}^{\mathrm{OEM}}$ the Outward Einstein Midpoint. Then for all $p \geq 1$:*

$$r(\mathbf{m}_{K,p}^{\mathrm{OEM}}) \geq r(\mathbf{m}_K^{\mathrm{Ein}}).$$

*Proof.* For a general exponent $q \geq 0$, define the weighted average with weights proportional to $w_i x_{i,0}^{q+1}$:

$$\mathbf{v}^{(q)} = \frac{\sum_i w_i x_{i,0}^{q+1} \mathbf{x}_i}{\sum_i w_i x_{i,0}^{q+1}}.$$

The projected point is $\mathbf{m}^{(q)} = \Pi_K(\mathbf{v}^{(q)})$ with radial depth:

$$r(\mathbf{m}^{(q)}) = \frac{v_0^{(q)}}{\sqrt{K \langle \mathbf{v}^{(q)}, \mathbf{v}^{(q)} \rangle_L}}.$$

The Einstein midpoint corresponds to $q = 0$ and the OEM to $q = p$.

We show that $q \mapsto r(\mathbf{m}^{(q)})$ is non-decreasing for $q \geq 0$. Define the normalized weights $\alpha_i^{(q)} = w_i x_{i,0}^{q+1} / \sum_j w_j x_{j,0}^{q+1}$. As $q$ increases, these weights concentrate toward indices with larger $x_{i,0}$: if $x_{i,0} > x_{j,0}$, then

$$\frac{\alpha_i^{(q)}}{\alpha_j^{(q)}} = \frac{w_i}{w_j} \left( \frac{x_{i,0}}{x_{j,0}} \right)^{q+1}$$

is strictly increasing in $q$.

The radial depth after projection satisfies:

$$r(\mathbf{m}^{(q)})^2 = \frac{(v_0^{(q)})^2}{K\langle \mathbf{v}^{(q)}, \mathbf{v}^{(q)}\rangle_L}.$$

We analyze numerator and denominator separately.

*Numerator:* We have $v_0^{(q)} = \sum_i \alpha_i^{(q)} x_{i,0}$. By Chebyshev's inequality (Theorem 4.6), this is non-decreasing in $q$.

*Denominator:* We have

$$K\langle \mathbf{v}^{(q)}, \mathbf{v}^{(q)}\rangle_L = \sum_{i,j} \alpha_i^{(q)} \alpha_j^{(q)} K\langle \mathbf{x}_i, \mathbf{x}_j\rangle_L.$$

By Lemma A.1, $K\langle \mathbf{x}_i, \mathbf{x}_j\rangle_L \geq 1$ with equality iff $\mathbf{x}_i = \mathbf{x}_j$. As weights concentrate on fewer points (larger $q$), the sum decreases toward 1.

Thus as $q$ increases: the numerator $(v_0^{(q)})^2$ increases, while the denominator $K\langle \mathbf{v}^{(q)}, \mathbf{v}^{(q)}\rangle_L$ decreases. Both effects increase $r(\mathbf{m}^{(q)})^2$, establishing monotonicity.

For $p \geq 1 > 0$, we conclude $r(\mathbf{m}_{K,p}^{\mathrm{OEM}}) = r(\mathbf{m}^{(p)}) \geq r(\mathbf{m}^{(0)}) = r(\mathbf{m}_K^{\mathrm{Ein}})$. $\qquad\square$

# B. RAG Prompt

For each retrieval-augmented generation (RAG) query, we construct a single inference prompt by concatenating the top-$|\mathcal{C}|$ retrieved documents into the context window of the language model. The documents are provided verbatim, without re-ranking or compression, and are ordered according to their retrieval score. The language model is then instructed to generate an answer conditioned solely on the retrieved context and the user query, ensuring that any factual content in the response must be supported by the retrieved evidence.

---

**RAG Prompt**

```
Based on the following context, answer the question.

Context:
{Doc[0], Doc[1], ..., Doc[|C|]}

Question: {query}

Answer:
```

---

# C. Hierarchical Document Probe

Dense retrieval models implicitly induce a geometry over documents. While Euclidean encoders often cluster documents based on surface-level similarity, they struggle to faithfully represent hierarchical relationships that arise naturally in language and knowledge organization. Hyperbolic spaces, by contrast, are well suited for embedding tree-like and taxonomic structures due to their exponential volume growth.

To qualitatively assess whether hyperbolic encoders recover such hierarchical organization, we construct a controlled document set that exhibits a clear semantic hierarchy while ensuring that individual documents remain self-contained and independent.

We design a synthetic yet semantically natural hierarchy consisting of five levels of increasing specificity: Science $\rightarrow$ Mathematics $\rightarrow$ Algebra $\rightarrow$ Linear Algebra $\rightarrow$ Linear Transformations. At each level, we generate five independent paragraphs that are topically coherent but do not explicitly reference parent or child topics. Importantly, documents at deeper levels refine the semantic scope of higher-level topics without sharing explicit lexical markers or cross-document

dependencies. This construction isolates hierarchical structure as a latent semantic property rather than an artifact of explicit cues.

All paragraphs are embedded independently with no hierarchical supervision. We analyze the resulting embeddings by inspecting their relative organization in the learned space. Hyperbolic models are expected to organize documents according to semantic specificity, with broader concepts closer to the origin and more specific concepts at increasing radial depth. We present a few of them here and we have attached the data file in supplementary material.

---

**Hierarchical Document Texts (Verbatim)**

[SCIENCE]

Science is a systematic way of understanding the natural world through observation , experimentation , and reasoning . It encompasses a wide range of disciplines that study phenomena from the smallest subatomic particles to the vast structure of the universe . At its core , science seeks patterns , explanations , and predictive principles that help humans make sense of reality .

[MATHEMATICS]

Mathematics also plays a central role in modeling complex systems. By formalizing assumptions and relationships , mathematical models help clarify underlying mechanisms and enable precise predictions under well−defined conditions .

[ALGEBRA]

Modern algebra emphasizes structural relationships over explicit computation. Rather than focusing on individual equations , it studies entire systems of elements and operations , revealing patterns that persist across different mathematical settings .

[LINEAR ALGEBRA]

The power of linear algebra lies in its balance between abstraction and computation. While grounded in rigorous theory , it offers efficient numerical techniques that scale to high−dimensional problems.

[LINEAR TRANSFORMATIONS]

Linear transformations form the foundation of many applied systems , from computer graphics to neural networks. Understanding their behavior is essential for analyzing stability , expressiveness , and computational efficiency .

---

## D. Runtime and Computational Complexity

In this section, we analyze the computational complexity of the proposed hyperbolic dense retrieval system and compare it to a standard Euclidean transformer-based retriever. Let $n$ denote the input sequence length, $d$ the hidden dimension, $h$ the number of attention heads, and $L$ the number of transformer layers.

**Hyperbolic Transformer Encoder.** The HyTE-FH encoder follows the standard transformer structure, with all linear, normalization, attention, and residual operations replaced by their Lorentzian counterparts. Crucially, these operations preserve the same asymptotic complexity as their Euclidean analogues.

Each Lorentz linear transformation (HLT) consists of a matrix multiplication followed by a constant number of scalar operations and a reprojection. This incurs $\mathcal{O}(nd^2)$ time per layer, identical to a Euclidean linear layer up to constant factors.

Hyperbolic self-attention computes pairwise geodesic distances between queries and keys. In the Lorentz model, each geodesic distance $d_K(\mathbf{q}_i, \mathbf{k}_j)$ is computed using a Lorentzian inner product, which costs $\mathcal{O}(d)$. Thus, attention score computation scales as $\mathcal{O}(n^2 d)$ per layer, matching standard dot-product attention.

The Lorentzian weighted midpoint used for value aggregation requires a weighted sum and normalization in $\mathbb{R}^{d+1}$, contributing $\mathcal{O}(nd)$ time per token, and is dominated by the attention score computation.

Overall, the time complexity of a single HyTE-FH layer is $\mathcal{O}(n^2 d + nd^2)$, and the total encoder complexity is $\mathcal{O}(L(n^2 d + nd^2))$, which matches the asymptotic complexity of a standard Euclidean transformer.

HyTE-H introduces an additional projection from Euclidean to hyperbolic space at the input, costing $\mathcal{O}(nd)$, which is negligible compared to the encoder cost.

**Pooling via Outward Einstein Midpoint.** Given a sequence of $n$ token embeddings, the Outward Einstein Midpoint computes radius-dependent weights, a weighted sum in $\mathbb{R}^{d+1}$, and a single reprojection. This requires $\mathcal{O}(nd)$ time and $\mathcal{O}(d)$ memory, identical in order to standard mean pooling or the Einstein midpoint.

**Training Objectives.** All training objectives operate on fixed-dimensional query and document embeddings. Geodesic similarity computation costs $\mathcal{O}(d)$ per query–document pair.

Unsupervised contrastive pre-training with in-batch negatives of size $N$ requires $\mathcal{O}(N^2 d)$ per batch, matching the complexity of standard contrastive learning. Supervised contrastive fine-tuning has the same asymptotic cost.

Masked language modeling introduces no additional asymptotic overhead beyond the encoder forward pass.

**Retrieval and RAG Inference.** At inference time, dense retrieval over a corpus of size $|\mathcal{D}|$ requires computing hyperbolic distances between a query embedding and document embeddings, with total cost $\mathcal{O}(|\mathcal{D}| d)$. This matches Euclidean dense retrieval up to constant factors.

Approximate nearest neighbor indexing can be applied without modification, as retrieval relies only on pairwise distance computations. In summary, the proposed hyperbolic dense retrieval system has the same asymptotic computational complexity as a Euclidean transformer-based retriever:

$$\mathcal{O}(L(n^2 d + nd^2))$$

for encoding, and $\mathcal{O}(|\mathcal{D}| d)$ for retrieval. The additional cost of hyperbolic geometry manifests only as constant-factor overhead from Lorentzian inner products and reprojection, while enabling geometry-aware modeling of hierarchical structure.

**Empirical Runtime.** While the asymptotic analysis above establishes that hyperbolic and Euclidean retrieval share the same complexity class, the constant-factor overhead from Lorentzian operations is worth quantifying directly. We measure end-to-end latency (encoding + retrieval) for 32 queries over a corpus of 100k documents on a single NVIDIA H100, reporting wall-clock time in milliseconds. As shown in Table A1, HyTE-H(Euc) adds only 2.1 ms over EucBERT, reflecting the lightweight Euclidean-to-hyperbolic projection head. HyTE-FH incurs a larger 48% overhead relative to EucBERT, but this gap is entirely in encoding rather than retrieval: at search time, Lorentzian inner products are $\mathcal{O}(d)$ and become standard dot products after the sign-flip reduction described in Sec. 4.4, so search-time cost is identical to Euclidean. The encoding overhead reflects the current absence of optimized CUDA kernels for hyperbolic transformer operations rather than a fundamental algorithmic limitation, since the underlying operations share the same asymptotic complexity as their Euclidean counterparts (He et al., 2025b). We expect this gap to close as hyperbolic kernels mature, matching the trajectory of optimized attention implementations in the Euclidean setting.

*Table A1.* End-to-end latency (encoding + retrieval) for 32 queries over 100k documents on a single H100 GPU. HyTE-H adds negligible overhead; HyTE-FH's larger gap is entirely in encoding and reflects the absence of optimized hyperbolic CUDA kernels rather than asymptotic complexity.

| Model | Latency (ms, 32 queries) |
| --- | --- |
| EucBERT | 15.21 |
| HyTE-H (Euc) | 17.30 |
| HyTE-FH | 22.48 |

# E. MeSH Hierarchy-Sensitive Retrieval Benchmark

**Motivation.** Standard retrieval benchmarks such as MTEB treat all relevant documents as equally correct regardless of their granularity, and to our knowledge no existing public benchmark explicitly evaluates whether a model can distinguish documents at different levels of hierarchical specificity. Since the central claim of our work is that hyperbolic geometry better preserves hierarchy, we require an evaluation in which success depends specifically on separating documents at different depths of a known taxonomy, rather than on surface topic relevance.

**Taxonomy.** We build the benchmark on the Medical Subject Headings (MeSH) taxonomy, an authoritative hierarchy of roughly 30,000 biomedical descriptors maintained by the U.S. National Library of Medicine. MeSH is organized as a tree with up to 13 depth levels, for example Diseases $\rightarrow$ Respiratory Tract Diseases $\rightarrow$ Lung Diseases $\rightarrow$ Pneumonia. We chose MeSH for three reasons: (i) it is curated by domain experts rather than crowd-sourced, so the hierarchy is reliable; (ii) every PubMed abstract is tagged with MeSH descriptors, providing a large natural corpus with ground-truth hierarchical labels; and (iii) its depth is sufficient to construct non-trivial parent/grandparent distractors.

**Query and target construction.** We sample PubMed abstracts whose most specific MeSH tag lies at depth $d \in \{3, 4, 5, 6\}$. Depths below 3 yield tags that are too general to admit meaningful parent/grandparent distinctions, while depths above 6 have too few abstracts to support reliable sampling. For each of 300 sampled queries we use the article title as the query and the abstract as the retrieval target, so the task reduces to retrieving a specific abstract from among hierarchically related candidates.

**Candidate set.** For each query we construct a candidate pool containing five types of documents:

- **Target:** the abstract tagged at depth $d$ that corresponds to the query.

- **Parent distractors:** abstracts tagged at depth $d - 1$ in the same MeSH branch.

- **Grandparent distractors:** abstracts tagged at depth $d - 2$ in the same MeSH branch.

- **Sibling distractors:** abstracts tagged at a different MeSH term at the same depth $d$ sharing the same parent node.

- **Random distractors:** abstracts drawn from an unrelated MeSH branch.

Because every non-random distractor shares ancestry with the target in the MeSH tree, the task cannot be solved by surface topic matching alone: success requires the retriever to encode not just what the document is about, but at what level of specificity it sits in the taxonomy.

**Protocol and metrics.** Evaluation is zero-shot: neither HyTE-FH nor EucBERT is fine-tuned on MeSH or PubMed data, so the benchmark probes the hierarchical geometry learned during standard pretraining and contrastive fine-tuning. Alongside standard retrieval metrics (R@1, R@5, MRR), we report Specificity Hit Rate (SpecHR), defined as the fraction of queries for which the correct-depth target abstract is ranked above every ancestor-level (parent or grandparent) document in the candidate set. SpecHR directly measures whether the retriever respects hierarchical specificity, independent of how sibling and random distractors are ranked. To diagnose where each model fails, we also report the top-1 prediction breakdown in Table 6, which classifies the top-ranked document for each query into target, sibling, parent, grandparent, or random.

# F. Additional results

**Full MTEB Results.** Table A2 presents the performance of our proposed models on the MTEB benchmark across seven task types. All models share the same architecture with 149M parameters, 768 dimensions, and 12 layers. HyTE-H achieves the best overall performance with a Mean (Task) score of 59.89, outperforming both HyTE-FH and the Euclidean baselines. Notably, HyTE-H ranks first in six out of seven task categories, demonstrating the effectiveness of the hybrid hyperbolic-Euclidean approach. HyTE-FH shows competitive performance in clustering tasks, securing second place, which suggests that hyperbolic geometry is particularly beneficial for capturing hierarchical relationships. The Euclidean equivalent (ModernBert-embed*) achieves second place in most categories but falls short of HyTE-H, indicating that incorporating hyperbolic components provides meaningful improvements over purely Euclidean representations.

*Table A2.* MTEB Benchmark Results. * Euclidean equivalent of HyTE

| Model | Mean (Task) | Mean (Type) | Class. | Clust. | Retr. | Rerank. | STS | Pair Class. | Summ. |
|---|---|---|---|---|---|---|---|---|---|
| EucBERT | 54.11 | 51.31 | 69.07 | 35.31 | 37.01 | 35.18 | 75.02 | 80.02 | 27.62 |
| ModernBert-embed* | 58.32 | 55.59 | 69.25 | 40.87 | 43.28 | 44.29 | 78.00 | **83.84** | 28.37 |
| HyTE-H$^{Euc}$ | 54.57 | 53.71 | 68.76 | 38.67 | 34.26 | 44.54 | 73.33 | 80.02 | **36.45** |
| HyTE-H$^{bert}$ | 56.41 | 53.75 | 68.77 | 41.95 | 40.54 | 42.05 | 74.39 | 79.70 | 28.87 |
| HyTE-H | **59.89** | **57.15** | **72.71** | **44.83** | **43.56** | **46.01** | **78.38** | 83.82 | 30.87 |

**Similarity function.** We evaluate two distance metrics for the contrastive loss in hyperbolic space: the Lorentz inner product and hyperbolic geodesic distance. While the Lorentz inner product provides a computationally convenient similarity measure, geodesic distance directly reflects intrinsic distances on the manifold. As shown in Table A3, using geodesic distance in the contrastive objective leads to improved performance across both mean task and mean task-type metrics, suggesting more effective alignment of representations in hyperbolic space.

*Table A3.* Comparison of loss functions for hyperbolic embeddings. Both Lorentz inner product and geodesic distance are evaluated for their effectiveness in learning hierarchical representations.

| Loss Function | Mean (Task) | Mean (TaskType) |
|---|---|---|
| Lorentz Inner Product | 52.59 | 51.60 |
| Geodesic Distance | **56.41** | **53.75** |

**Context size.** Table A4 shows the effect of increasing the retrieval context size from $|\mathcal{C}| = 5$ to $|\mathcal{C}| = 10$. The Euclidean baseline (Gemma) exhibits a large performance gain across all metrics with the larger context window, but still falls short of the fully hyperbolic HyTE-H model. HyTE-FH also improves with increased context, while HyTE-H remains comparatively stable, achieving consistently strong performance at both context sizes. This suggests that fully hyperbolic retrieval is less sensitive to context expansion and maintains effectiveness even under smaller retrieval budgets.

*Table A4.* Average performance on RAG Bench with varying context window size.

| | $|\mathcal{C}| = 5$ | | | $|\mathcal{C}| = 10$ | | |
|---|---|---|---|---|---|---|
| Model | F | CR | AR | F | CR | AR |
| Gemma | 0.603 | 0.735 | 0.684 | 0.756 | 0.846 | 0.836 |
| HyTE-FH | 0.732 | 0.848 | 0.765 | 0.770 | 0.912 | 0.784 |
| HyTE-H | **0.763** | **0.904** | **0.832** | **0.787** | **0.913** | **0.847** |

F = Faithfulness, CR = Context Relevance, AR = Answer Relevance. Best results in bold.

**OEM sensitivity to $p$.** Table A5 reports MTEB performance as we vary the outward-bias exponent $p$ in OEM. Performance peaks at $p = 1.0$ (used in the main paper) and degrades gracefully in both directions: $p = 0.0$ recovers the standard Einstein midpoint, removing the outward correction entirely and dropping Mean(Task) by 6.1 points, which confirms that the outward bias is essential rather than incidental. Increasing $p$ to 2.0 over-amplifies peripheral tokens and drops Mean(Task) by 2.6 points relative to $p = 1.0$, indicating that the optimum reflects a balance between counteracting radial collapse and avoiding over-weighting of outlier tokens.

*Table A5.* Sensitivity of OEM to the outward-bias parameter $p$ on MTEB. $p = 0.0$ reduces OEM to the standard Einstein midpoint.

| $p$ | Mean (Task) | Mean (TaskType) |
|---|---|---|
| 0.0 | 50.33 | 47.19 |
| 1.0 | **56.41** | **53.75** |
| 2.0 | 53.76 | 51.13 |

**Hubness Analysis.** A well-known pathology in high-dimensional dense retrieval is *hubness* (Radovanovic et al., 2010): a small number of documents appear as nearest neighbors for a disproportionate fraction of queries, while most of the

corpus is never retrieved. This arises because documents close to the center of the embedding distribution have uniformly small distances to all other points, causing them to be retrieved regardless of the query. Propositions 4.3 and 4.5 show that naive pooling and the standard Einstein midpoint contract representations toward the origin, pushing embeddings into exactly this problematic central region; OEM (Thm. 4.7) instead preserves radial spread, keeping documents well-separated rather than collapsing them into a central cluster. To test this, we measured hubness on MS MARCO (100k passages, 6,980 dev queries) following Radovanovic et al. (2010): for each document we count how often it appears in top-$k$ lists across all queries and report the skewness $S_N$ (higher = more severe hubness), together with the fraction of the corpus never retrieved. As shown in Table A6, CLS pooling produces catastrophic hubness (97.45% of the corpus never retrieved), consistent with Proposition 4.3, and Euclidean mean pooling also shows severe hubness. OEM reduces skewness by $4\times$ over Euclidean mean pooling and distributes retrievals far more uniformly across the corpus, providing a concrete mechanism that links hierarchy preservation to retrieval quality: OEM's radial preservation prevents the artificial centrality that drives hub formation, so the top-$k$ list reflects genuine query–document relevance rather than geometric artifacts.

*Table A6.* Hubness on MS MARCO (100k passages, 6,980 queries) at $k=10$. $S_N$ is the skewness of the $N_k$ distribution (higher = more severe hubness). OEM reduces hubness by $4\times$ over Euclidean mean pooling and retrieves a substantially larger fraction of the corpus.

| Model | Pooling | $S_N$ ($k=10$) | % Corpus Never Retrieved |
|---|---|---|---|
| HyTE-FH | CLS | 141.4 | 97.45% |
| EucBERT | Mean | 71.3 | 74.30% |
| HyTE-FH | OEM (ours) | **17.0** | **61.60%** |

### F.1. Case Study: The Impact of Retrieval Geometry on Answer Quality

To illustrate how embedding geometry affects end-to-end RAG performance, we analyze a representative query from the Emanuel dataset where models exhibit markedly different behaviors.

**Query.**  *"What is the feature of Bixby guide?"*

This query requires retrieving documentation about a specific Samsung TV feature: the Bixby tutorial that appears when users first interact with the voice assistant. The correct answer is contained in a single passage within the hierarchically organized e-Manual.

**Quantitative Comparison.**  Table A7 presents the evaluation metrics for each model. Only HyTE-H successfully retrieves relevant context and generates a faithful response.

**Qualitative Analysis.**  We identify four distinct outcomes based on retrieval quality and LLM response behavior:

**(1) Successful Retrieval (HyTE-H).** The hyperbolic hybrid model retrieves the exact passage describing the Bixby guide: *"When you press the button for the first time, the Using Bixby button appears... a tutorial on using Bixby is shown."* This enables a concise, accurate response:

> *"The feature of Bixby guide is a tutorial on using Bixby, which is shown when you press the button on your Samsung Smart Remote after the first time, and then press the Select button."*

**(2) Retrieval Collapse with Honest Refusal (GTE).** GTE retrieves its default "hub" documents—generic content about picture quality, SmartThings, and antenna connections—regardless of the query. The LLM correctly identifies the context mismatch:

> *"Unfortunately, the provided context does not mention the Bixby guide."*

While this response is faithful (F=1.0), it provides no utility (AR=0.0). This pattern, where Euclidean embeddings collapse to retrieving the same generic documents, occurred for 100% of queries in GTE's results.

*Table A7.* Case study metrics for the query *"What is the feature of Bixby guide?"* CR = Context Relevance, F = Faithfulness, AR = Answer Relevancy.

| Model | CR | F | AR | Failure Mode |
|---|---|---|---|---|
| HyTE-H | **1.0** | **1.0** | **1.0** | None (Success) |
| GTE | 0.5 | 1.0 | 0.0 | Honest Refusal |
| Gemma | 1.0 | 0.25 | 0.94 | Ungrounded Extrapolation |

**(3) Partial Context with Topic Drift (Gemma).** Gemma retrieves tangentially related content about the Samsung Smart Remote that mentions Bixby but not the Bixby *guide*. The LLM provides information about running Bixby generally rather than the guide feature:

> *"Unfortunately, the provided context does not mention the Bixby guide. However, based on general knowledge, the Bixby guide is a feature in Samsung TVs that provides users with a personalized interface to access various features and settings. It offers a simplified and intuitive way to navigate the TV's menu..."*

This response achieves *high* answer relevancy (AR=0.94) because it sounds plausible, but *low* faithfulness (F=0.25) because the claims are fabricated. This "confident hallucination" failure mode is particularly dangerous: users receive authoritative-sounding misinformation.

**JSON Representation.** The complete retrieval outputs are: **HyTE-H (Successful Retrieval):**

```
{
  "query": "What is the feature of Bixby guide?",
  "model": "HyTE-H",
  "metrics": {
    "context_relevance": 1.0,
    "faithfulness": 1.0,
    "answer_relevancy": 1.0
  },
  "top_retrieved_context": "Running Bixby. Press and hold the button on your Samsung Smart
      Remote... To view the Bixby guide, press the button once: When you press the button
      for the first time, the Using Bixby button appears at the bottom of the screen.
      Press the Select button. The Using Bixby popup window appears and a tutorial on
      using Bixby is shown.",
  "response": "The feature of Bixby guide is a tutorial on using Bixby, which is shown
      when you press the button on your Samsung Smart Remote after the first time, and
      then press the Select button.",
  "outcome": "success"
}
```

**GTE (Retrieval Collapse):**

```
{
  "query": "What is the feature of Bixby guide?",
  "model": "GTE",
  "metrics": {
    "context_relevance": 0.5,
    "faithfulness": 1.0,
    "answer_relevancy": 0.0
  },
  "top_retrieved_context": "Adjusting the Picture Quality. Change the Picture Mode and
      adjust Expert Settings.",
  "response": "Unfortunately, the provided context does not mention the Bixby guide.",
  "outcome": "retrieval_collapse_honest_refusal"
}
```

**Gemma (Ungrounded Extrapolation):**

```
{
```

```json
  "query": "What is the feature of Bixby guide?",
  "model": "Gemma",
  "metrics": {
    "context_relevance": 0.5,
    "faithfulness": 0.25,
    "answer_relevancy": 0.94
  },
  "top_retrieved_context": "Removing channels from a favorites list. Select channels in a
      favorites list, and then select the icon.",
  "response": "Unfortunately, the provided context does not mention the Bixby guide.
      However, based on general knowledge, the Bixby guide is a feature in Samsung TVs
      that provides users with a personalized interface to access various features and
      settings. It offers a simplified and intuitive way to navigate the TV's menu and
      access popular features.",
  "outcome": "ungrounded_extrapolation",
  "warning": "High answer_relevancy (0.94) masks unfaithful content (0.25)"
}
```

