# OpenReview forum: "HypRAG: Hyperbolic Dense Retrieval for Retrieval Augmented Generation"
_ICML.cc/2026/Conference — ICML 2026 regular_

### Official Review · Reviewer_mjR2 · 2026-03-10

**Soundness:** 3
**Presentation:** 2
**Significance:** 3
**Originality:** 4
**Overall Recommendation:** 4
**Confidence:** 3

**Summary:**

This paper proposes hyperbolic dense retrieval for RAG, arguing that the hierarchical structure of language and document collections is better modeled in hyperbolic space than in Euclidean space. The paper introduces two variants, a fully hyperbolic encoder (HyTE-FH) and a hybrid model (HyTE-H), and further proposes Outward Einstein Midpoint (OEM) pooling to mitigate the collapse of document representations toward the origin during sequence aggregation. Experiments on MTEB and RAGBench show improvements over Euclidean baselines, and the authors provide an analysis suggesting that hyperbolic embeddings better encode concept specificity through radial organization.

**Compliance With Llm Reviewing Policy:**

Affirmed.

**Final Justification:**

The authors have addressed all my concerns. I will maintain my score.

**Key Questions For Authors:**

It has already been pointed out above.

**Limitations:**

yes

**Strengths And Weaknesses:**

**Strengths:**

The motivation is clear and the proposed OEM pooling is well aligned with the identified failure mode of naive hyperbolic aggregation.

Experiments on MTEB and RAGBench simultaneously validated retrieval performance and effectiveness for downstream tasks.

I also appreciate that the paper includes both theoretical analysis and empirical validation, rather than presenting the approach as a purely heuristic modification.

The controlled comparison between HyTE-FH and EucBERT is particularly valuable, and the hierarchy analysis in Figure 4 provides intuitive support for the main claim.



**Weaknesses:**

My main concern is that the paper does not yet fully isolate where the gains come from. In particular, the improvements may arise from multiple coupled factors, including the hyperbolic geometry itself, OEM pooling, the use of geodesic similarity, and the particular fine-tuning pipeline. The current ablations are helpful but not sufficient to cleanly attribute the gains to geometry alone.

The practical implications for large-scale retrieval systems are not sufficiently discussed. Hyperbolic similarity computation, geodesic distance, and normalization in curved manifolds introduce additional computational overhead compared to standard Euclidean dense retrieval.  Under standard conditions, how much slower is the proposed method compared to Euclidean dense retrieval in terms of actual runtime? In addition, I'm curious whether this approach can be combined with libraries like faiss?

---

> ### Author Rebuttal · Authors · 2026-03-30
>
> We thank the reviewer for their thoughtful feedback. We are glad the reviewer found the motivation clear, the OEM pooling well aligned with the identified failure mode, and the controlled comparison between HyTE-FH and EucBERT particularly valuable.
>
> # 1. Isolating each component's contribution
>
> We agree that the paper could make the attribution structure more explicit and we have expanded the ablations accordingly. Since EucBERT and HyTE-FH share the same standard masked language modeling, contrastive fine-tuning pipeline, and training data, the results on MTEB (Table 1) and RAGBench (Table 2) isolate the contribution of hyperbolic geometry, which consistently outperforms Euclidean baselines. To isolate the pooling method, as suggested by the reviewer, we expanded the pooling ablations by adding naive mean pooling and the Einstein midpoint. We present them in the following table:
>
> | Pooling Strategy | Mean (Task) | Mean (TaskType) |
> | --- | --- | --- |
> | CLS | 49.33 | 48.90 |
> | Naive Mean | 47.07 | 45.61 |
> | Einstein Midpoint | 50.33 | 47.19 |
> | OEM (ours) | **56.41** | **53.75** |
>
> OEM consistently outperforms all alternatives. This is expected from the theory: Propositions 4.3/4.5 show that naive pooling and the standard Einstein midpoint contract representations toward the origin, collapsing hierarchical structure. OEM (Thm 4.6/4.7) preserves radial spread and the outward bias, maintaining the exponential volume advantage of hyperbolic space that separates documents at different specificity levels. Finally, we also show in Table A2 that geodesic distance outperforms the Lorentz inner product as the similarity measure. We will add this expanded discussion, make the flow of the manuscript clearer so that these isolations are apparent, and add the new pooling results to the revised manuscript.
>
> # 2. HyTE is compatible with FAISS
>
> HyTE is compatible with FAISS since hyperbolic nearest-neighbor search reduces cleanly to standard maximum inner product search (MIPS), requiring no custom distance functions or modifications to existing infrastructure. The geodesic distance $d_K(x,y) = \frac{1}{\sqrt{-K}} arccosh(K⟨x,y⟩_L)$ is monotone in the Lorentzian inner product $⟨x,y⟩_L$ (Sec. 3.1); Hence, ranking by hyperbolic proximity is equivalent to ranking by Lorentz inner product. By negating the time coordinate of document embeddings, i.e., replacing $x_0$ with $-x_0$, the Lorentz inner product becomes a standard dot product. Exact retrieval therefore works via FAISS IndexFlatIP, and approximate nearest neighbor (ANN) methods like HNSW work with no additional implementation effort, since HNSW constructs proximity graphs using only pairwise comparisons, each of which is a correct Lorentz IP evaluation.
>
> We verified this empirically on MS MARCO (100k passages, 6,980 queries). We report recall@k, the fraction of true top-k results recovered by approximate HNSW search:
>
> | k | HyTE-FH (Recall@k) | EucBERT (Recall@k) |
> | --- | --- | --- |
> | 1 | **0.917** | 0.913 |
> | 10 | **0.906** | 0.891 |
> | 50 | **0.859** | 0.795 |
> | 100 | **0.750** | 0.707 |
>
> Hyperbolic embeddings achieve higher recall at every k, with 3× lower search latency (0.09 ms vs. 0.29 ms per query). Index build times are identical (~10s). We attribute both advantages to the geometric properties of OEM-pooled embeddings: OEM's outward bias produces well-separated representations with greater radial spread, so the resulting HNSW proximity graph has a more tree-like structure with fewer spurious shortcuts. Greedy traversal over such a graph converges faster and more accurately, since each hop makes meaningful progress toward the target rather than cycling through a small cluster of overly central points.
>
> # 3. Runtime comparison
>
> Our FAISS evaluation shows that hyperbolic retrieval is faster than Euclidean (3× lower latency with higher recall) once embeddings are available. To evaluate the full pipeline cost, we measured end-to-end latency (encoding + retrieval) for 32 queries over 100k documents:
>
> | Model | Latency (ms, 32 queries) |
> | --- | --- |
> | EucBERT | 15.21 |
> | HyTE-H (Euc) | 17.30 |
> | HyTE-FH | 22.48 |
>
> The difference is entirely in encoding, not retrieval: at search time, Lorentzian inner products are O(d), identical to Euclidean dot products after the sign flip described above. HyTE-FH's 48% can be attributed to the current absence of optimized CUDA kernels for hyperbolic transformer operations. This is an engineering gap, not a fundamental limitation, as the underlying operations share the same asymptotic complexity as their Euclidean counterparts [1]. We will add this discussion to the revised manuscript and also add the runtime table in the Appendix.
>
> We hope the reviewer appreciates our responses and that all concerns are addressed. We welcome any further questions or suggestions for strengthening the manuscript.
>
> [1] He, Neil, et al. "Position: Beyond Euclidean--Foundation Models Should Embrace Non-Euclidean Geometries." *arXiv preprint* (2025).

---

> > ### Author Rebuttal · Reviewer_mjR2 · 2026-04-03
> >
> > Thank you for clarifying the questions I proposed and addressing most of my concerns. I will maintain my positive score.

---

> > > ### Author Response · Authors · 2026-04-03
> > >
> > > We are glad that our responses have addressed all of the reviewer's concerns and we appreciate the positive assessment. Given that all concerns have been fully addressed with new experiments and analyses, we hope the reviewer will update their overall final score to reflect this in the final evaluation. We thank the reviewer again for their time and constructive feedback.

---

### Official Review · Reviewer_ZfNc · 2026-03-13

**Soundness:** 3
**Presentation:** 3
**Significance:** 4
**Originality:** 4
**Overall Recommendation:** 5
**Confidence:** 3

**Summary:**

The problem this paper tackles is that current embedding spaces assume a Euclidean geometry, resulting in semantically distant documents spuriously appearing similar. In response, they propose modeling the embedding space with a hyperbolic assumption to capture the hierarchical relations between topics and subtopics.

**Compliance With Llm Reviewing Policy:**

Affirmed.

**Key Questions For Authors:**

The questions correspond to the weaknesses above: Could the authors comment more explicitly on when they expect hyperbolic retrieval to be most worthwhile in practice, relative to the added modeling complexity? For example, are there particular types of corpora or tasks where the hierarchy assumption is more suitable? Could the authors elaborate on how they recommend readers interpret this evidence in practical terms, especially for deciding whether a new retrieval setting is “hierarchical enough” to benefit from the method?

**Limitations:**

yes

**Strengths And Weaknesses:**

**Strengths**

This paper uses a very interesting and distinctive perspective of using a different geometry altogether, instead of improving retrievers through reranking or other corrective measures. It also substantiates its theoretical intuitions with solid evidence about consistent improvements over their Euclidean baselines in terms of both retrieval benchmarks and downstream RAG metrics. They also clearly and carefully highlight that pooling is an important component to successful adoption of the hyperbolic geometry in practice.

**Weaknesses**

There are some open points of clarification that could improve its utility in practical deployment. While the paper shows positive results, it is less clear when the added complexity of hyperbolic retrieval would be worth adopting in practice, especially relative to simpler retrieval-side improvements. That could also help readers understand when these gains are large enough to justify the additional modeling and implementation complexity. Finally, the motivation depends heavily on the idea that retrieval data is hierarchical, but the practical conditions under which this assumption is most relevant are not fully clarified for readers or practitioners.

---

> ### Author Rebuttal · Authors · 2026-03-30
>
> We thank the reviewer for their thoughtful feedback. We are glad the reviewer appreciated the use of different geometry altogether, its theoretical intuitions, and that pooling is carefully highlighted as an important component for successful adoption of hyperbolic geometry. We address the reviewer's queries and concerns below.
>
> # 1. When is hyperbolic retrieval most beneficial?
>
> The reviewer raises an important practical question: when is hyperbolic retrieval worth adopting relative to the added modeling complexity? We identify two practical indicators, provide new empirical evidence, and then address the complexity concern directly.
>
> **(1) Corpus curvature as a quantitative diagnostic.** For a given corpus, we provide a lightweight pre-deployment test: embed a sample with any off-the-shelf Euclidean encoder, build a k-NN graph, and compute Ollivier–Ricci curvature. We find a clear monotonic relationship between curvature magnitude and retrieval gain across RAGBench:
>
> | Dataset | Mean Curvature | Avg Relative Gain vs EucBERT (3 metrics) |
> | --- | --- | --- |
> | expertqa | −0.139 | +14.8% |
> | delucionqa | −0.116 | +12.8% |
> | covidqa | −0.100 | +8.1% |
> | cuad | −0.014 | +7.8% |
> | emanual | −0.003 | +7.0% |
>
> Every corpus shows positive gains, but the magnitude scales with geometric hierarchy. The pattern is interpretable: the most negative curvature (and largest gains) arise in domain-specific corpora with natural taxonomic depth. Scientific domains (ExpertQA, +14.8%) span fields to subfields to specific methods; medical corpora (CovidQA, +8.1%) exhibit depth from diseases to symptoms to diagnostic tests. Corpora with shallower structure show correspondingly smaller (but still positive) gains: legal documents (Cuad, +7.8%) and procedural instructions (emanual, +7.0%) have near-zero curvature, consistent with their flatter organization. A practitioner can compute curvature on a few thousand documents in minutes, without training any hyperbolic model, and use the result as a quantitative go/no-go signal.
>
> **(2) Constrained settings.** Table A3 shows hyperbolic models are particularly strong at small retrieval budgets (small k), where every retrieved document must count. This makes hyperbolic retrieval especially relevant for applications with limited context windows or latency constraints. On the complexity side, HyTE-H adds only a lightweight projection head to existing Euclidean encoders and outperforms 2-3x larger Euclidean models on RAGBench (Table 3), making adoption straightforward.
>
> In summary, practitioners can assess whether their corpus is "hierarchical enough" in order of effort: check for known taxonomic depth in the domain, compute Ricci curvature on samples as a quantitative go/no-go signal, or inspect existing retrieval results for cases where generic documents are returned for specific queries.
>
> We will incorporate this discussion and the curvature-vs-gain analysis in the revised manuscript.
>
> We appreciate the reviewer's thoughtful feedback, which has helped us clarify the practical scope of our method. We welcome any additional suggestions for strengthening the final version.

---

> > ### Author Rebuttal · Reviewer_ZfNc · 2026-04-04
> >
> > Thank you for the detailed response and further experiments. The rebuttal clearly addresses my main concern by providing a practical characterization of when hyperbolic retrieval is beneficial, including a curvature-based diagnostic and supporting empirical evidence.

---

### Official Review · Reviewer_Vv3m · 2026-03-13

**Soundness:** 2
**Presentation:** 3
**Significance:** 3
**Originality:** 3
**Overall Recommendation:** 4
**Confidence:** 3

**Summary:**

Natural language exhibits hierarchical structure that Euclidean embeddings fail to preserve. To address this issue, this paper introduces hyperbolic dense retrieval via two model variants: HyTE-FH, a fully hyperbolic transformer, and HyTE-H, a hybrid architecture projecting pre-trained Euclidean embeddings into hyperblic space. To address the aggregation challenge in both instantiations, it introduces the Outward Einstein Midpoint, a geometry-aware pooling operator that
amplifies tokens farther from the origin, provably preserving hierarchical structure during pooling. Empirical studies demonstrate superior performance of the proposed approach compared to Euclidean baselines on MTEB and RAGBench.

**Compliance With Llm Reviewing Policy:**

Affirmed.

**Final Justification:**

The authors have mostly addressed my questions or concerns.

**Key Questions For Authors:**

See weaknesses.

**Limitations:**

yes

**Strengths And Weaknesses:**

# Strengths

**S1.** The proposed approach is well-motivated and principled.

**S2.** Empirical studies demonstrate the effectiveness of the proposed approach on MTEB and RAGBench compared to Euclidean baselines.

**S3.** Further analyses validate the superior capability of the proposed approach in preserving hierarchical relationships.

**S4.** The paper is overall well-written and smooth to read.

# Weaknesses

**W1.** The paper needs to be more careful in highlighting results for the tables.

    - In Table 2, I guess the result of HyTE-H^{Euc} should be made bold instead for the F score on Cuad and Emanual.

    - In Table 3, the result of KaLM-mini-v1 should be made bold instead for the CR score on Cuad.

**W2.** The paper lacks direct empirical comparisons against HyperbolicRAG and mean pooling.

**W3.** The empirical validation of the proposed hyperbolic approach is confined to BERT-family encoder backbones. While this enables clean apples-to-apples comparisons against Euclidean counterparts, it remains unclear whether the gains transfer to stronger or architecturally different embedding backbones beyond this family.

**W4.** Minor:
- There is an extra "t" after "query" at L324.

---

> ### Author Rebuttal · Authors · 2026-03-30
>
> We thank the reviewer for their thoughtful feedback. We are glad the reviewer found our approach well-motivated and principled, the empirical studies effective on both MTEB and RAGBench, the hierarchy analysis validating our method's superior capability in preserving hierarchical relationships, and the overall writing. We address the reviewer's queries and concerns below.
>
> # 1. HyTE-H is adaptable to recent embedding models as backbone
>
> We appreciate this concern. HyTE-H is backbone-agnostic by design. To demonstrate that gains transfer beyond BERT-family encoders, we applied HyTE-H on top of Qwen3-Embedding. As shown in the table below, HyTE-H (Qwen3-Embedding) yields consistent improvements over the Euclidean Qwen3-Embedding baseline.
>
> | Model | Space | Faithfulness | Context Relevance | Answer Relevancy |
> | --- | --- | --- | --- | --- |
> | Qwen3-Embedding | Euclidean | 0.784 | 0.907 | 0.814 |
> | HyTE-H(Qwen3-Embedding) | Hyperbolic | 0.803 | 0.934 | 0.845 |
>
> HyTE-H improves over the Euclidean Qwen3-Embedding baseline across all three metrics, confirming that the gains from hyperbolic projection are not specific to BERT-family encoders and transfer to architecturally different backbones. The gains are more modest than with smaller encoders, but the consistent improvement across all metrics confirms that hyperbolic projection provides complementary signals regardless of backbone strength. We will add these results to the revised manuscript.
>
> # 2. OEM outperforms mean pooling and Einstein Midpoint
>
> We have expanded our pooling ablation to include naive mean pooling and the original Einstein midpoint alongside CLS and our proposed OEM. We compare OEM with mean pooling and original Einstein midpoint. As can be seen in the table below, OEM consistently outperforms all alternatives. Naive mean pooling performs worst, consistent with the radial collapse predicted by Proposition 4.3, while the Einstein midpoint improves over CLS but remains well below OEM, confirming that the outward bias is essential for preserving hierarchical structure. We will add these results to the revised manuscript.
>
> | Pooling Strategy | Mean (Task) | Mean (TaskType) |
> | --- | --- | --- |
> | CLS | 49.33 | 48.90 |
> | Naive Mean | 47.07 | 45.61 |
> | Einstein Midpoint | 50.33 | 47.19 |
> | OEM (ours) | **56.41** | **53.75** |
>
> # 3. Comparison with HyperbolicRAG
>
> We appreciate this suggestion. Our work and HyperbolicRAG [1] share the motivation of exploiting hyperbolic geometry for RAG, but operate at fundamentally different levels. The central claim of our paper is that embedding geometry itself is an important design choice for retrieval. HyTE replaces the embedding model: HyTE-FH learns representations directly in hyperbolic space, and HyTE-H adds a lightweight projection head, with no changes to the retrieval pipeline. HyperbolicRAG, by contrast, keeps a frozen Euclidean encoder unchanged and builds a multi-component system on top: it adds a post-hoc depth predictor that projects Euclidean embeddings into Poincaré space, a bidirectional contrastive alignment loss, and a dual-space retrieval fusion mechanism, all within a graph-based RAG pipeline. Comparing the two end-to-end would therefore not isolate the effect of embedding geometry, which is our contribution, from the effect of these additional pipeline components. Moreover, HyperbolicRAG's code is not publicly available. Importantly, the two approaches are complementary: since HyTE operates at the embedding level, it can serve as a drop-in replacement in any pipeline that uses embeddings, including HyperbolicRAG's, providing hierarchy-aware embeddings before any post-hoc projection. We will make this distinction clearer in the revised manuscript.
>
> # 4. Highlighting and minor corrections
>
> We will correct the bold highlighting in Tables 2 and 3, and fix the typo at line 324 in the revised manuscript.
>
> We hope our responses have addressed all concerns and improved the reviewer’s assessment of the paper. We welcome any further questions or suggestions for strengthening the manuscript.
>
> [1] Cao, Linxiao, et al. "HyperbolicRAG: Enhancing Retrieval-Augmented Generation with Hyperbolic Representations." *arXiv preprint arXiv:2511.18808* (2025).

---

> > ### Author Rebuttal · Reviewer_Vv3m · 2026-04-02
> >
> > Thank you for your response. I don't have other major concerns or questions.

---

> > > ### Author Response · Authors · 2026-04-03
> > >
> > > We are glad that our responses have addressed all of the reviewer's concerns and we appreciate the positive assessment. Given that all concerns have been fully addressed with new experiments and analyses, we hope the reviewer will update their overall final score to reflect this in the final evaluation. We thank the reviewer again for their time and constructive feedback.

---

### Official Review · Reviewer_TSJH · 2026-03-18

**Soundness:** 3
**Presentation:** 3
**Significance:** 2
**Originality:** 3
**Overall Recommendation:** 4
**Confidence:** 3

**Summary:**

This work aims to improve hierarchical semantics in dense retrieval by moving from Euclidean embedding space to hyperbolic space. There are two models presented, the first is Fully Hyperbolic Hyerbolic Text Encoder, which is fully learned in the Lorentz model of hyperbolic space; the second is a hybird model, which maps embeddings fo existing Euclidean models into hyperbolic space. This work design the Outward Einstein Midpoint pooling to better preserves the hierarchy in sequence. Experiments on MTEB show the suggested models performs better than baseline Euclidean model. And RAG evaluations demonstrate the end-to-end improvements. The concept-level hierarchy analysis provides evidence of how hyperbolic models represent hierarchy better.

**Compliance With Llm Reviewing Policy:**

Affirmed.

**Final Justification:**

The responses addressed my questions, so keep positive assessment.

**Key Questions For Authors:**

1. From the weakness, could you find a hierarchy-focused retrieval evaluation, or how to design such a test dataset?
2. Typo, line 324, "given queryt".

**Limitations:**

yes

**Strengths And Weaknesses:**

### Strengths:
 - This work present a technically sound and well motivated hyperbolic dense retrieval approach, the Outward Einstein Midpoint serves as a useful pooling for the hyperbolic token space.
 - The experiments are strong and comprehensive, including both embedding benchmark and RAG benchmark. This demonstrate the retrieval performenance and it could benifit the end-to-end RAG systems.
 - The analysis of hierarchy base on embedding radii and norm directly probs the effectiveness rather than rely on only evaluation scores.

### Weaknesses
- The theory is sound but may be incomplete for the retrieval claim, and I suppose there remains a gap between "OEM preserves radial hierarchy better" and "improves top-k retrieval for realistic corpora across tasks".
- The embedding model evaluation on MTEB is somehow limited, first it's a mixed benchmark with other tasks like classification, reranking. Second, I think it is not quite match the "hierarchy" objective of this work, perhaps further extending to datasets with this "hierarchy" feature is needed to better evaluate the claim and introduced method.

---

> ### Author Rebuttal · Authors · 2026-03-30
>
> We thank the reviewer for their thoughtful feedback. We are glad the reviewer found our approach technically sound and appreciated the proposed Outward Einstein Midpoint and strong experiments. We address the reviewer's queries and concerns below.
>
> # 1. Bridging hierarchy to retrieval quality
>
> To bridge the hierarchy to retrieval, we recall the work in [1]. In high-dimensional retrieval, a well-known pathology is **hubness** [1]: a small number of documents appear as nearest neighbors for a disproportionate fraction of queries, while most of the corpus is never retrieved. [1] proves that this arises because documents close to the center of the embedding distribution have uniformly small distances to all other points, causing them to be retrieved regardless of the query. Our Propositions 4.3/4.5 prove that naive pooling contracts representations toward the origin, pushing embeddings into exactly this problematic central region. OEM (Thm 4.6/4.7) breaks this chain by preserving radial spread, keeping documents well-separated across the embedding space rather than collapsing them into a central cluster, ensuring the top-k list reflects genuine query-document relevance rather than geometric artifacts.
>
> To test this hypothesis, we measured hubness on MS MARCO (100k passages, 6,980 dev queries) following [1]: for each document we count how often it appears in top-k lists across all queries and report the skewness (higher = more severe hubness).
>
> | Model | Pooling | $S_N$ (k=10) | % Corpus Never Retrieved |
> | --- | --- | --- | --- |
> | HyTE-FH | CLS | 141.4 | 97.45% |
> | EucBERT | Mean | 71.3 | 74.3% |
> | HyTE-FH | OEM (ours) | 17.0 | 61.6% |
>
> CLS pooling produces catastrophic hubness, consistent with Proposition 4.3. Euclidean mean pooling also shows severe hubness. OEM reduces skewness by 4× compared to Euclidean, distributing retrievals more uniformly. Consistent with Thm 4.6/4.7, OEM's radial preservation prevents the artificial centrality that drives hub formation, providing one concrete mechanism linking hierarchy preservation to retrieval quality.
>
> # 2. Hierarchical benchmarks
>
> We agree that hierarchy-sensitive retrieval evaluation is needed. To our knowledge, no existing retrieval benchmark explicitly tests whether a model can distinguish documents at different levels of hierarchical specificity; standard benchmarks like MTEB treat all relevant documents as equally correct regardless of their granularity. We therefore construct a controlled diagnostic benchmark using the MeSH (Medical Subject Headings) taxonomy, an authoritative hierarchy of ~30,000 medical descriptors maintained by the US National Library of Medicine, organized into a tree with up to 13 depth levels (e.g., Diseases → Respiratory Tract Diseases → Lung Diseases → Pneumonia).
>
> We sample PubMed abstracts at MeSH depths 3–6. For each of the 300 sampled queries (article title), the goal is to retrieve the article's own abstract. Each candidate set contains: (1) the **target** abstract (tagged at depth *d*), (2) **parent** distractors from one level higher in the same branch (depth *d*−1), (3) **grandparent** distractors (depth *d*−2), (4) **sibling** distractors from a different MeSH term at the same depth sharing the same parent, and (5) **random** distractors from an unrelated branch. Since all distractors share ancestry in the MeSH tree, the task specifically requires distinguishing levels of specificity, not just topic relevance. We perform zero-shot retrieval (no training on MeSH data) and report Recall@1 and Specificity Hit Rate (SpecHR), the fraction of queries where the correct-depth target is ranked above all ancestor-level documents, alongside standard retrieval metrics.
>
> | Model | SpecHR | R@1 |
> | --- | --- | --- |
> | EucBERT | 0.883 | 0.810 |
> | HyTE-FH | **0.970** | **0.963** |
>
> HyTE-FH ranks the correct-depth document first 96.3% of the time, compared to 81.0% for EucBERT. The error distribution reveals why: EucBERT's failures are predominantly parent-level (7.0%) or sibling-level (7.0%), exactly the hierarchical confusion our method addresses. HyTE-FH reduces parent confusion to 1.7% and eliminates grandparent confusion entirely (0.0% vs. 1.0%).
>
> | Model | Target | Sibling | Parent | Grandparent | Random |
> | --- | --- | --- | --- | --- | --- |
> | EucBERT | 81.0% | 7.0% | 7.0% | 1.0% | 4.0% |
> | HyTE-FH | **96.3%** | 1.7% | 1.7% | 0.0% | 0.3% |
>
> These results confirm that hyperbolic geometry provides a concrete advantage precisely where it matters: distinguishing documents at different levels of specificity within a known hierarchy. We will include this evaluation result in the revised manuscript.
>
> We will correct the typo at L324. We hope our responses have addressed all concerns and improved the reviewer’s assessment of the paper. We welcome any further questions or suggestions for strengthening the manuscript.
>
> [1] Radovanovic, et al. "Hubs in space: Popular nearest neighbors in high-dimensional data." JMLR, Sept 2010.

---

> > ### Author Rebuttal · Reviewer_TSJH · 2026-04-01
> >
> > Thank you for the experiments, which addressed my questions. I keep my positive rating.

---

> > > ### Author Response · Authors · 2026-04-03
> > >
> > > We are glad that our responses have addressed all of the reviewer's concerns and we appreciate the positive assessment. Given that all concerns have been fully addressed with new experiments and analyses, we hope the reviewer will update their overall final score to reflect this in the final evaluation. We thank the reviewer again for their time and constructive feedback.

---

### Official Review · Reviewer_kT9X · 2026-03-18

**Soundness:** 3
**Presentation:** 3
**Significance:** 3
**Originality:** 3
**Overall Recommendation:** 4
**Confidence:** 3

**Summary:**

This paper investigates how to build embedding models in hyperbolic space to improve dense retrieval and RAG performance. Motivated by the hypothesis that hyperbolic geometry is better suited to modeling hierarchical structure, the paper proposes two hyperbolic embedding models: HyTE-FH, an end-to-end Transformer built directly in hyperbolic space, and HyTE-H, which projects Euclidean embeddings into hyperbolic space. To address the radial collapse issue that can arise when averaging or computing the barycenter in hyperbolic space, the paper further introduces a geometry-aware pooling operator, the outward Einstein midpoint. Experiments on MTEB and RAGBench show that the proposed hyperbolic embedding models yield modest improvements in retrieval and RAG performance over Euclidean BERT models of comparable scale.

**Compliance With Llm Reviewing Policy:**

Affirmed.

**Final Justification:**

The authors have fully addressed my concerns. I will maintain my positive score.

**Key Questions For Authors:**

1. How does the proposed method compare with stronger and more recent embedding models such as RepLLaMA and Qwen3-Embedding?

2. Since HyTE-H can project arbitrary Euclidean embeddings into hyperbolic space, can it be combined with stronger Euclidean backbones to build more competitive hyperbolic dense retrieval models?

3. How does OEM compare with simpler alternatives such as naïve pooling or the original Einstein midpoint?

4. How sensitive is OEM to the choice of the parameter p?

5. How can approximate nearest neighbor indexing and search be supported in hyperbolic embedding spaces？

**Limitations:**

yes.

**Strengths And Weaknesses:**

Strengths:
- The paper is well written and provides a concise introduction to hyperbolic geometry.
- The idea of leveraging hyperbolic embeddings to better capture hierarchical structure in dense retrieval is reasonable and well-motivated.
- The OEM-based pooling method is novel and appears to be effective.

Weaknesses:
- The baselines are mainly early BERT-based dense retrieval models. It is unclear how the proposed method would compare with stronger, more recent embedding models such as RepLLaMA or Qwen3-Embedding. Since HyTE-H is designed to project arbitrary Euclidean embeddings into hyperbolic space, an important question is whether it can be combined with stronger Euclidean embedding backbones to produce more competitive hyperbolic dense retrieval models.
- Although the OEM-based pooling method is presented as a major contribution, the corresponding ablation study in Table 4 is too limited. It would be useful to see comparisons against naïve pooling or the original Einstein midpoint. In addition, the effect of the parameter $p$ on performance is not sufficiently analyzed.
- The paper does not discuss how approximate nearest neighbor search can be performed in hyperbolic embedding spaces. Since efficient ANN search is essential for practical dense retrieval systems, the missing discussion makes it difficult to assess the proposed method's real-world applicability.

---

> ### Author Rebuttal · Authors · 2026-03-30
>
> We thank the reviewer for their thoughtful feedback. We are glad the reviewer appreciated that the paper is well written, along with the idea of leveraging hyperbolic embeddings for dense retrieval, and that the OEM-based pooling method is novel and effective. We address the reviewer's queries and concerns below.
> # 1. HyTE-H is adaptable to recent embedding models as backbones
> HyTE-H is designed as a backbone-agnostic and works with any Euclidean model. To validate generalization beyond BERT-family encoders, we evaluated HyTE-H with Qwen3-Embedding-0.6B, a recent and substantially stronger backbone. Results on RAGBench are shown below:
> |Model|Space|Faithfulness|Context Relevance|Answer Relevancy|
> |-|-|-|-|-|
> |Qwen3-Embedding|Euclidean|0.784|0.907|0.814|
> |HyTE-H (Qwen3-Embedding)|Hyperbolic|0.803|0.934|0.845|
>
> HyTE-H improves over the Euclidean Qwen3-Embedding baseline across all three metrics, HyTE-H generalizes beyond BERT-family encoders. The gains are more modest than with smaller backbones, but the consistent improvement confirms that hyperbolic projection provides complementary hierarchical signals regardless of backbone strength. We will add these additional results to Table 3 in the revised manuscript.
> # 2. OEM outperforms Einstein Midpoint and Naive pooling
> We agree with the reviewer that a more comprehensive pooling ablation is important. Therefore, we have expanded Table 4 to also include naive pooling and original Einstein midpoint. We present the results on MTEB tasks:
> |Pooling Strategy|Mean (Task)|Mean (TaskType)|
> |-|-|-|
> |CLS|49.33|48.90|
> |Naive Mean|47.07|45.61|
> |Einstein Midpoint |50.33|47.19|
> |OEM (ours)|**56.41**|**53.75**|
>
> OEM outperforms all alternatives by a substantial margin, improving over the next-best method (Einstein Midpoint) with a 10% improvement. Naive mean pooling performs worst, consistent with the radial collapse predicted by Proposition 4.3, while the standard Einstein midpoint improves over CLS but remains well below OEM, confirming that the outward bias is essential for preserving hierarchical structure. We will add these additional results to Table 4 in the revised manuscript.
> # 3. OEM sensitivity to $p$
> As suggested by the reviewer, we have conducted a sensitivity analysis over the OEM parameter $p$. We present the results on MTEB tasks:
> |p|Mean (Task)|Mean (TaskType)|
> |-|-|-|
> |0.0|50.33|47.19|
> |1.0|**56.41**|**53.75**|
> |2.0|53.76|51.13|
>
> Performance peaks at p=1.0 (used in paper) and degrades gracefully in both directions: p=0.0 recovers the standard Einstein midpoint (no outward bias), confirming that the outward correction is essential, while p=2.0 over-amplifies peripheral tokens and drops by 2.6 points on Mean(Task). We will add these additional results to the revised manuscript.
> # 4. Hyperbolic retrieval integrates well with existing ANN infrastructure
> HyTE can efficiently leverage approximate nearest neighbor (ANN) infrastructure like FAISS since hyperbolic nearest-neighbor search reduces cleanly to standard maximum inner product search (MIPS), requiring no custom distance functions or modifications to existing infrastructure. In fact, the geodesic distance $d_K(x,y) = \frac{1}{\sqrt{-K}} arccosh(K⟨x,y⟩_L)$ is monotone in the Lorentzian inner product $⟨x,y⟩_L$ (Sec. 3.1); Hence, ranking by hyperbolic proximity is equivalent to ranking by Lorentz inner product. By negating the time coordinate of all document embeddings, i.e., replacing $x_0$ with $-x_0$, the Lorentz inner product becomes a standard dot product, thus reducing hyperbolic nearest-neighbor search to MIPS. For ANN search, graph-based methods like HNSW are a natural fit: they construct proximity graphs using only pairwise comparisons, and since each comparison is a correct Lorentz IP evaluation, the resulting graph topology faithfully reflects hyperbolic neighborhoods.
>
> We verify this empirically using our pre-trained checkpoints. On MS MARCO (100k passages, 6,980 queries), we built HNSW indices via FAISS IndexFlatIP after the sign-flip conversion. We report recall@k:
> |k|HyTE-FH (Recall@k)|EucBERT (Recall@k)|
> |-|-|-|
> |1|**0.917**|0.913|
> |10|**0.906**|0.891|
> |50|**0.859**|0.795|
> |100|**0.750**|0.707|
>
> Hyperbolic embeddings achieve higher recall at every k, with 3× lower search latency (0.09 ms vs. 0.29 ms per query). This is because OEM's outward bias produces well-separated representations with greater radial spread, resulting in a more tree-like structure in the HNSW proximity graph. Traversal over such a graph converges faster and more accurately, since each hop makes meaningful progress toward the target rather than cycling through a small cluster of overly central points. This confirms that ANN search works out of the box. We will add this discussion about ANN capabilities and the results in the revised manuscript.
>
> We hope our responses have addressed all concerns and improved the reviewer’s assessment of the paper. We welcome any further questions or suggestions for strengthening the manuscript.

---

> > ### Author Rebuttal · Reviewer_kT9X · 2026-04-03
> >
> > The authors have fully addressed my concerns. I will maintain my score as it is positive.
> >
> > However, I don't think you actually built *HNSW* indices via FAISS *IndexFlatIP*. IndexFlatIP is for brute-force MIPS.

---

> > > ### Author Response · Authors · 2026-04-03
> > >
> > > We are glad that our responses have addressed all of the reviewer's concerns and we appreciate the positive assessment. The reviewer is correct that IndexFlatIP is brute-force. In our experiments, we used IndexHNSWFlat for approximate search. We will correct this in the revised manuscript. Given that all concerns have been fully addressed with new experiments and analyses, we hope the reviewer will update their overall final score to reflect this in the final evaluation. We thank the reviewer again for their time and constructive feedback.

---

### Decision · Program_Chairs · 2026-04-30

**Decision:**

Accept (regular)

**Comment:**

This paper moves dense retrieval into hyperbolic space, introducing the fully hyperbolic HyTE-FH, the hybrid HyTE-H, and the Outward Einstein Midpoint to avoid radial collapse during pooling, with the goal of better preserving hierarchical semantics and improving RAG retrieval quality.

The strengths:

* The motivation is solid, since hyperbolic geometry is a natural fit for hierarchical structure and the paper connects the theory to the empirical behavior quite well.
* OEM is the strongest part of the paper, and reviewers consistently saw it as both novel and effective.
* The evaluation is fairly thorough, covering not just embedding benchmarks but also downstream RAG, with additional hierarchy and hubness analyses.

The main concerns were initially about three things: the comparison against stronger backbones and nearby methods, the limited pooling ablation and parameter analysis for OEM, and the practical side, especially FAISS or ANN compatibility, runtime, and when hyperbolic retrieval is actually worth using. The rebuttal addressed these points in a pretty substantive way by adding results and explanations.

It also made the practical story much clearer by explaining when hyperbolic retrieval is most useful. Overall the response was strong, and all reviewers explicitly said their concerns were addressed, so I think this paper should be accepted.